# Prefrontal cortex molecular clock modulates development of depression-like phenotype and rapid antidepressant response in mice

David H. Sarrazin [1,11], Wilf Gardner[1,2,11], Carole Marchese[1,2], Martin Balzinger[1,2], Chockalingam Ramanathan [3], Marion Schott [1], Stanislav Rozov[4,5], Maxime Veleanu[6], Stefan Vestring[6,7], Claus Normann [6,8], Tomi Rantamäki [4,5], Benedicte Antoine[9], Michel Barrot[1,2], Etienne Challet [1], Patrice Bourgin [1,10] & Tsvetan Serchov [1,2,6] ✉

Depression is associated with dysregulated circadian rhythms, but the role of intrinsic clocks in mood-controlling brain regions remains poorly understood. We found increased circadian negative loop and decreased positive clock regulators expression in the medial prefrontal cortex (mPFC) of a mouse model of depression, and a subsequent clock countermodulation by the rapid antidepressant ketamine. Selective *Bmal1*KO in CaMK2a excitatory neurons revealed that the functional mPFC clock is an essential factor for the development of a depression-like phenotype and ketamine effects. *Per2* silencing in mPFC produced antidepressant-like effects, while REV-ERB agonism enhanced the depression-like phenotype and suppressed ketamine action. Pharmacological potentiation of clock positive modulator ROR elicited antidepressant-like effects, upregulating plasticity protein Homer1a, synaptic AMPA receptors expression and plasticity-related slow wave activity specifically in the mPFC. Our data demonstrate a critical role for mPFC molecular clock in regulating depression-like behavior and the therapeutic potential of clock pharmacological manipulations influencing glutamatergic-dependent plasticity.

Major depressive disorder (MDD)[1] is one of the most common mental disorders and a leading cause of disability worldwide. Although beneficial, widely used monoamine-based antidepressants have delayed therapeutic effect, low response rate, and common adverse effects[2]. Even the robustly and rapidly acting antidepressant treatments of sleep deprivation (SD)[3] and subanesthetic dose ketamine[4] come with certain limitations, including short duration of effectiveness and presence of transient psychotomimetic effects. Different chronotherapies, strategies that modulate biological clocks and the sleep–wake cycle, including various SD and light paradigms, have been developed as alternative treatments for depression[5].

[1]Centre National de la Recherche Scientifique (CNRS), University of Strasbourg, Institute of Cellular and Integrative Neurosciences (INCI) UPR 3212, Strasbourg, France. [2]University of Strasbourg Institute for Advanced Study (USIAS), University of Strasbourg, Strasbourg, France. [3]Institute for Physiology I, University of Freiburg, Medical Faculty, Freiburg, Germany. [4]Laboratory of Neurotherapeutics, Drug Research Program, Division of Pharmacology and Pharmacotherapy, Faculty of Pharmacy, University of Helsinki, Helsinki, Finland. [5]SleepWell Research Program, Faculty of Medicine, University of Helsinki, Helsinki, Finland. [6]Department of Psychiatry and Psychotherapy, Medical Center - University of Freiburg, Faculty of Medicine, University of Freiburg, Freiburg, Germany. [7]Berta-Ottenstein-Programme for Clinician Scientists, Faculty of Medicine, University of Freiburg, Freiburg, Germany. [8]Center for Neuromodulation, Medical Center, Faculty of Medicine, University of Freiburg, Freiburg, Germany. [9]Sorbonne Université, INSERM, Centre de Recherches St-Antoine (CRSA), Paris, France. [10]CIRCSom (International Research Center for ChronoSomnology) & Sleep Disorders Center, Strasbourg University Hospital, Strasbourg, France. [11]These authors contributed equally: David H. Sarrazin, Wilf Gardner. ✉e-mail: serchov@inci-cnrs.unistra.fr

Circadian rhythms, physiological cycles of ~24 h, which influence a wide range of biochemical, physiological and behavioral processes, are often disturbed in mood disorders[6]. Likewise, majority of MDD patients show dysregulated sleep, including reduced slow wave sleep (SWS), potentially linked to altered mechanisms of plasticity in depression[7,8]. It has been suggested that molecular mechanisms of rapid antidepressants interact with and may be dependent on expression of both clock and synaptic plasticity genes[9–11]. Indeed, we have previously shown that the induction of the oscillating synaptic protein Homer1a in the medial prefrontal cortex (mPFC), a brain region strongly implicated in depression[12], is intricately involved in the antidepressant mechanisms of both SD and ketamine, and that it modulates glutamatergic signaling[13–15].

The intracellular oscillation of the circadian clock is driven by transcription/translation-based feedback/feedforward loops of a set of core clock genes: positive clock regulators BMAL1 and CLOCK rhythmically induce transcription of clock suppressor genes Period (*Per1*, *Per2*) and Cryptochrome (*Cry1*, *Cry2*), whose proteins in turn inhibit transactivation by BMAL1/CLOCK[16]. An additional loop including both activating and repressing regulatory factors is formed by the retinoic acid receptor-related orphan receptors (ROR α, β, γ) and the nuclear receptors REV-ERB (α, β)[17,18]. These receptors are considered among the primary molecular classes suitable for drug targeting[19].

The master circadian pacemaker, located in the suprachiasmatic nucleus (SCN), is indirectly linked to mood regulation[20], but disruption of the clock in other oscillating brain regions more closely implicated in depression may have a more direct role in the development of affective disorders[21–23]. Indeed, altered circadian gene expression is found in several extra-SCN brain regions of post-mortem MDD patients[24]. Thus, a better understanding of the molecular clock mechanisms associated with pathophysiology and therapeutic response might facilitate the identification of novel drug targets, improve treatment efficacy and lead to the development of preventive strategies[25].

Here, we provide evidence for specific alterations to the circadian clockwork in the mPFC of a mouse model of stress-induced depression-like phenotype, and subsequent clock modulation by the rapid antidepressant ketamine. Furthermore, we demonstrate pro- and antidepressant effects of different genetic and/or pharmacological manipulations of the molecular clock, showing effects on glutamatergic-dependent homeostatic plasticity in the mPFC, implicating a direct causative role for mPFC clock dysfunction in depression and demonstrating the therapeutic potential of such clock modulations.

## Results

### Repetitive swim stress alters circadian clock gene expression in mPFC

To investigate the effects of a depression-like phenotype on the diurnal pattern of core clock gene expression in the mPFC, mice were subjected to the chronic behavioral despair paradigm (CDM), a mouse model of depression in which passive immobility and anhedonia (Supplementary Fig. 1a) are induced by daily 10 min swim sessions for 5 consecutive days (induction phase)[13,15,26–28]. Brain samples of naive (control) and CDM mice were collected every 4 h over a 24 h period and mRNA expression was analyzed by qRT-PCR (Fig. 1a). All investigated canonical clock genes in the mPFC, except *Clock*, exhibited robust rhythmic pattern of expression in both groups (Fig. 1b and Supplementary Fig. 1c). Our analyses revealed that the CDM paradigm causes significant modulation in several core clock gene oscillations: increased amplitude of negative loop genes *Per2* and *Cry2*; reduced amplitude of *Rorα* and downregulated expression of *Rorβ* and shifted acrophase of *Per2, Cry2, Bmal1, Rev-erbα*, and *Rorα* (Fig. 1b, c and Supplementary Fig. 1b). CDM did not significantly alter the daily patterns of clock gene expression in the SCN (Supplementary Fig. 1e, f)

and the locomotor activity or period length either under 12:12 light/dark (LD) or constant darkness (DD) conditions (Supplementary Fig. 1d).

To further validate our findings in other animal models, we also collected and analyzed tissue from mice subjected to the learned helplessness (LH)[29] and chronic corticosterone (CORT)[30] models of depression. Our data demonstrate significant increase of the clock-negative loop genes *Per1* and *Rev-erbα* expression and decreased level of the positive clock regulators *Bmal1* and *Rorα* in the mPFC of LH mice. However, sucrose preference and mPFC clock gene expression of the CORT model were not significantly different from controls (Supplementary Fig. 1g, h).

These results indicate selective disruption of the circadian molecular clockwork in the mPFC after induction of a depressive-like phenotype, including enhanced expression of clock suppressors genes and decreased level of clock-positive regulators (Fig. 1d).

### Ketamine countermodulates CDM effects on mPFC clock

To examine the effect of rapid antidepressant ketamine on mPFC molecular clock, naive (control) and CDM mice were acutely i.p. injected with a subanesthetic dose of 3 mg/kg ketamine at ZT00 and brain tissue was harvested at 4 h intervals beginning from ZT06 on the day of treatment until ZT06 24 h later (Fig. 2a). mPFC expression of clock suppressors genes *Per1*, *Per2*, *Cry1*, *Cry2*, and *Rev-erbα* was downregulated by ketamine in both control and CDM mice at various ZTs (Fig. 2b, c), leading to a significant decrease of the amplitude of *Per2* and *Cry2* (Supplementary Fig. 2a). Ketamine also robustly shifted the phase of several clock genes, thus causing a normalization of *Bmal1* acrophase to match control level (Supplementary Fig. 2a). Moreover, ketamine markedly upregulated the expression of *Rorα* (Fig. 2b). Likewise, ketamine application at ZT12 in control mice decreased the expression of the clock suppressors *Cry1* and *Cry2* and increased the levels of the positive clock regulator *Rorα* (Supplementary Fig. 2d). In contrast, ketamine (at ZT00) had no significant effect on any analyzed core clock gene in the SCN and did not shift circadian locomotor activity at DD (Supplementary Fig. 2b, c).

In summary, the rapid antidepressant ketamine modulated the mPFC clockwork, causing lasting downregulation of clock suppressor gene expression, specific normalization of *Bmal1* acrophase, and potentiation of the clock-positive regulator *Rorα* (Fig. 2d).

### *Bmal1* knockout in the excitatory CaMK2a neurons of mPFC affects depression-like behavior and Homer1a expression

The core circadian core clock gene *Bmal1* is considered a necessary element of the functional molecular clock[31]. To elucidate the role of the circadian clock in the regulation of depression-like behavior, we specifically disrupted *Bmal1* expression in the excitatory glutamatergic neurons of the mPFC. To achieve this, we used AAV-mediated expression of Cre-recombinase or control EGFP, under the control of the calcium/calmodulin-dependent protein kinase II alpha (CAMK2a) promoter in *Bmal1*flox mice (Fig. 3a, b). Our analyses revealed at least 50% successful downregulation of *Bmal1* mRNA and protein levels in the mPFC of CaMK2a-Cre injected mice, corresponding to the relative proportion of the CaMK2a expressing cells in this region (Fig. 3c–e and Supplementary Fig. 3a, b). *Bmal1* ablation alone did neither affect depression-like behavior assessed during the baseline tail suspension test (TST) or nosepoke sucrose preference test (SPT) (Fig. 3g, h) nor influenced general locomotor and anxiety-related behavior measured in the IntelliCage and open-field test (OFT) (Supplementary Fig. 3c, d). To assess the importance of *Bmal1* for the development of depression-like phenotype, CaMK2a-*Bmal1*KO and control CaMK2a-EGFP mice were subjected to the CDM paradigm (Fig. 3b). Repetitive swim stress caused a gradual increase of the immobility time of the control CaMK2a-EGFP expressing mice during the induction phase (days 2–5), while CaMK2a-*Bmal1*KO mutants showed delayed response with

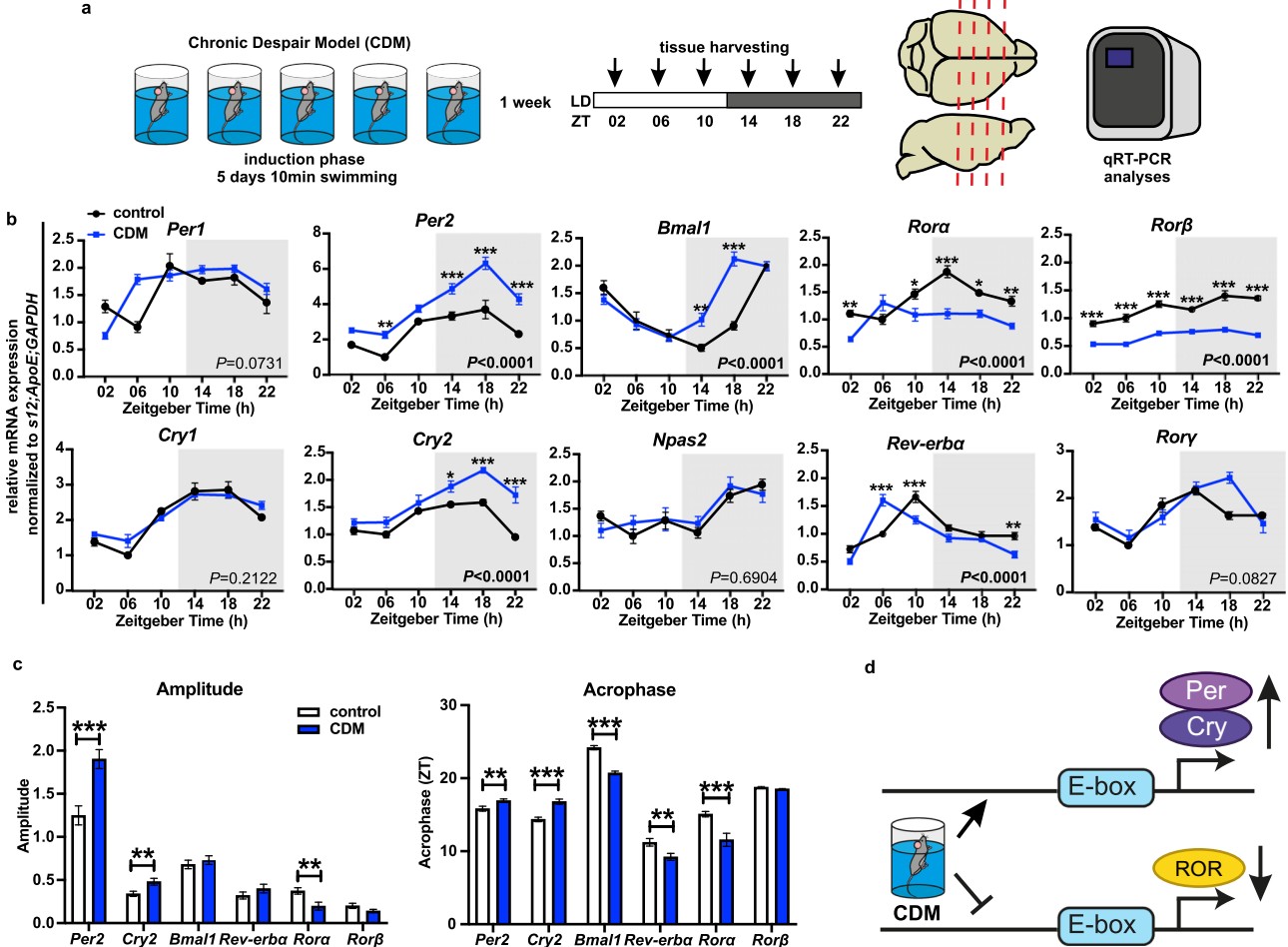

**Fig. 1 | Repetitive swim stress alters circadian clock gene expression in the mouse mPFC. a** Schematic overview of the experimental design: CDM paradigm (5 days of 10 min swimming), tissue harvesting time points and gene expression analyses. **b** Relative mRNA expression of clock genes *Per1, Per2, Cry1, Cry2, Bmal1, Npas2, Rorα, Rev-erbα, Rorβ, Rorγ* normalized to *s12, ApoE* and *GAPDH* in mPFC samples harvested from naive (control) and CDM mice every 4 h at 12:12 h LD condition (*n* = 5 mice per group, two-way ANOVA: *P* values of the CDM effect is displayed inside the graph; Bonferroni post hoc test: *$P$ < 0.05, **$P$ < 0.01, ***$P$ < 0.001 control vs. CDM). **c** Amplitude of rhythmic expression and acrophase (*n* = 30 mice: 5 ×6 ZT points; Extra sum of squares F test: **$P$ < 0.01, ***$P$ < 0.001). **d** Schema summarizing the effects of stress on clock gene expression in mPFC−CDM potentiates *Per2* and *Cry2* expression, while decreases Rors level. Data are presented as mean ± SEM. See also Supplementary Fig. 1 and Supplementary Data 1. Source data are provided as a Source Data file.

significantly increased immobility only on day 4 and day 5. Moreover, CaMK2a-*Bmal1*KO mice displayed decreased depression-like behavior during the CDM test phase compared to CaMK2a-EGFP mice (Fig. 3f). Consistently, the CDM increased the immobility time of the EGFP group in test phase TST in comparison to baseline, while it had no significant effect on *Bmal1*KO mice (Fig. 3g). Likewise, while both groups exhibited reduced motivation- and reward-oriented behavior in the SPT paradigm after CDM induction, CaMK2a-Cre expressing mice had significantly higher sucrose preference (Fig. 3h). In addition, *Bmal1*KO resulted in a marked downregulation of circadian genes from the negative clock regulatory loop, including *Per1, Per2*, and *Rev-erbα*, and a strong increase of *Rorα* expression in the mPFC of both naive (control) and CDM mice (Fig. 3i and Supplementary Fig. 3e).

A single ketamine injection (at ZT00) evoked an antidepressant response 24 h after application in the test phase TST and forced swim test (FST) (Fig. 3j) and decreased the expression of *Per1, Per2*, and *Rev-erbα* in the mPFC of CaMK2a-EGFP CDM mice (Fig. 3i and Supplementary Fig. 3e). However, the *Bmal1* deletion in mPFC CaMK2a neurons blocked the antidepressant effects of ketamine (Fig. 3j). Since a recent report implicated BMAL1 in the regulation of Homer1a[28], we also analyzed its expression in the mPFC. Indeed, the CDM-mediated Homer1a down-regulation, as well as Homer1a upregulation by ketamine, were ablated in CaMK2a-*Bmal1*KO animals (Fig. 3i and Supplementary Fig. 3f, g).

Taken together, our results suggest that *Bmal1* expression in excitatory CaMK2a neurons of the mPFC is an important factor in swim stress vulnerability and thus for the development of chronic depression-like phenotype, as well as playing a role in the anti-depressant effects of ketamine. Moreover, a functional molecular clock in mPFC, and particularly *Bmal1*, appears to be necessary in mediating Homer1a modulation by stress and ketamine (Fig. 3k).

## siRNA knockdown of Per2 in mPFC promotes antidepressant effect in CDM

Next, we investigated the role of *Per2* in the depressive-like phenotype via in vivo RNA expression silencing. After CDM induction, WT mice underwent bilateral stereotactic injection of either *Per2*-targeting Accell siRNA (siPer2) or non-targeting Accell siRNA (siCntr) in the mPFC[15] (Supplementary Fig. 4a). Behavioral tests were performed 48 h following injection, and tissue harvested immediately thereafter (at ZT06) (Fig. 4a). Our analysis revealed an approximately 40% decrease in *Per2* mRNA expression and 50% reduction in PER2 protein level in the mPFC of siPer2 mice (Fig. 4b, c). Reduced Per2 expression led to decreased immobility in both the TST and FST performed during the test phase (Fig. 4d, e), without significantly affecting locomotion or anxiety-like behavior in the OFT (Supplementary Fig. 4b). Further-more, Homer1a expression was increased in the mPFC of siPer2 mice

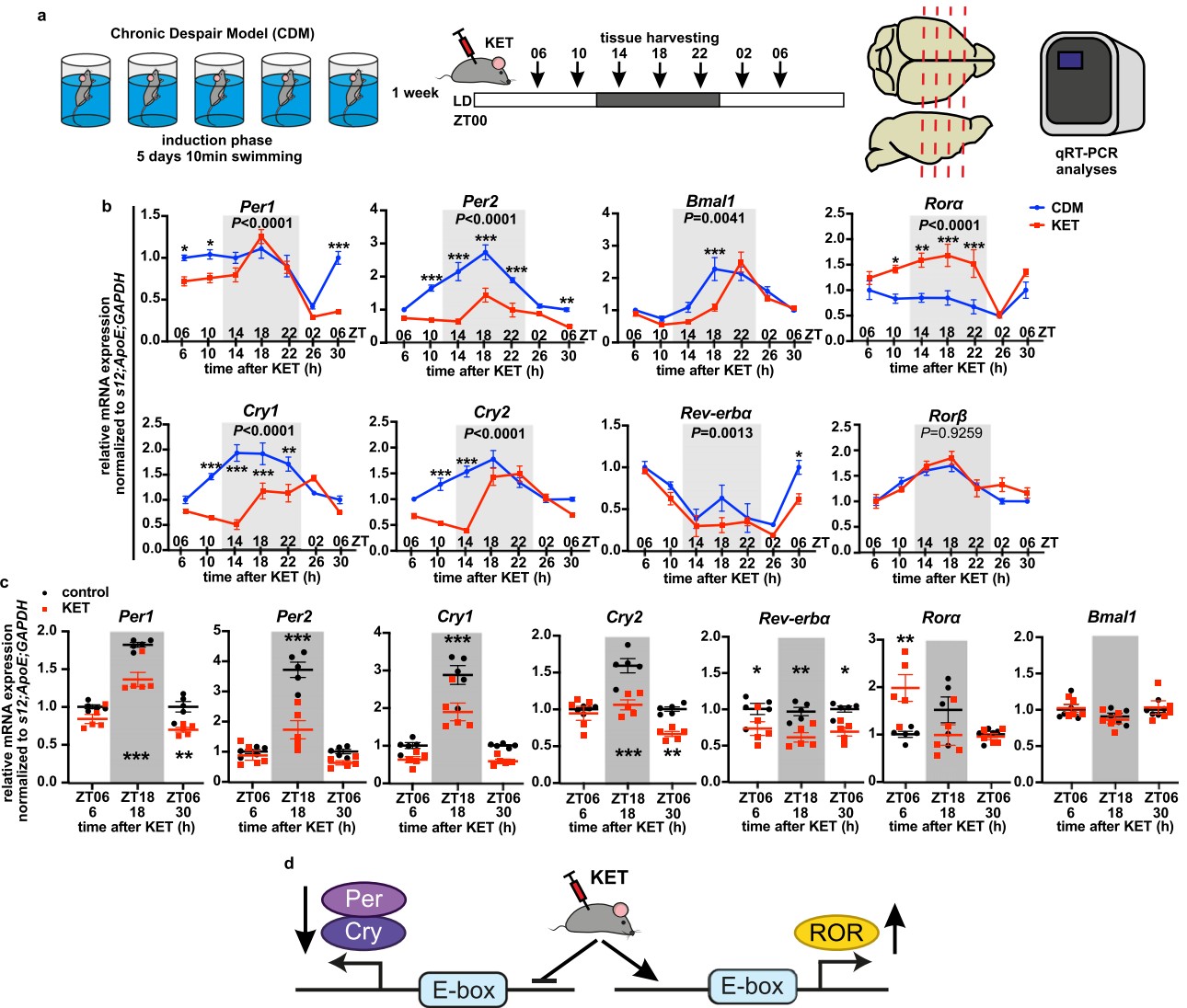

**Fig. 2 | Ketamine changes circadian clock gene expression in the mouse mPFC and opposes CDM effects. a** Schematic overview of the CDM paradigm, saline/3 mg/kg ketamine (KET) administration at ZT00 (start of the resting phase); tissue harvesting time points and gene expression analyses. **b** Relative mRNA expression of clock genes *Per1, Per2, Cry1, Cry2, Bmal1, Rorα, Rev-erbα, Rorβ* normalized to *s12, ApoE,* and *GAPDH* in mPFC samples from CDM mice injected at ZT00 with saline (CDM) or ketamine (KET) and harvested every 4 h from ZT06 (6 h after injection) till ZT06 (30 h post injection) (*n* = 5 mice per group, two-way ANOVA: *P* values of the CDM effect is displayed inside the graph, Bonferroni post hoc test: *P < 0.05, **P < 0.01, ***P < 0.001 CDM vs. KET). **c** Relative mRNA expression of clock genes

*Per1, Per2, Cry1, Cry2, Bmal1, Rorα, Rev-erbα, Rorβ* normalized to *s12, ApoE* and *GAPDH* in mPFC samples from naive mice injected at ZT00 with saline (control) and ketamine (KET) and harvested at ZT06 (6 h after injection), ZT18 (18 h after injection) and ZT06 (30 h post injection) (*n* = 5 mice per group, two-way ANOVA with Bonferroni post hoc test: *P < 0.05, **P < 0.01, ***P < 0.001 control vs. KET). **d** Schema summarizing the effects of ketamine on clock gene expression in mPFC – ketamine downregulates *Per* and *Cry* clock suppressors and increases *Rorα*. Data are presented as mean ± SEM. See also Supplementary Fig. 2 and Supplementary Data 1. Source data are provided as a Source Data file.

(Fig. 4f), while *Rev-erbα* expression was decreased (Supplementary Fig. 4c). These data suggest that knockdown of core clock repressor *Per2* expression in the mPFC produces an antidepressant-like effect coinciding with increased mPFC Homer1a expression (Fig. 4g).

### REV-ERBa modulates depression-like behavior through repression of the *Bmal1*-Homer1a axis in mPFC

REV-ERBα is the main transcriptional repressor of oscillating *Bmal1* expression and thus a modulator of circadian clockwork[17]. Notably, it is responsive to various molecules, allowing modulation of its activity[19,32–34]. To assess the role of REV-ERB activation in regulating depressive-like behavior, we utilized a specific synthetic REV-ERB agonist SR10067 (i.p., 30 mg/kg)[32]. Our data show that acute administration of SR10067 at ZT06 (at LD) causes transient decrease of the mouse locomotor and exploratory activity in IntelliCage during the

first 24 h post injection compared to vehicle (Supplementary Fig. 5a, b). In the absence of a stress protocol, acute or chronic 5 days injection of SR10067 at ZT06 had no significant effect on reward and motivation-oriented behavior assessed by the nosepoke SPT paradigm under LD conditions (Supplementary Fig. 5a, d, e). However, under DD, acute administration of SR10067 at CT06 significantly reduced sucrose preference 48 h after injection in comparison to vehicle (the mice were kept for 10 days under DD that caused anhedonic effect) and phase-shifted the locomotor activity (Supplementary Fig. 5a, c, d). We then tested the effects of single SR10067 application after the baseline swim on day 1 of the CDM paradigm under LD (Fig. 5a). Throughout CDM induction, SR10067-treated mice exhibit consistently elevated immobility time compared to vehicle-treated controls. This effect was maintained in the test phase FST and TST (Fig. 5b, c). Similarly, in the SPT, SR10067-treated mice exhibited more

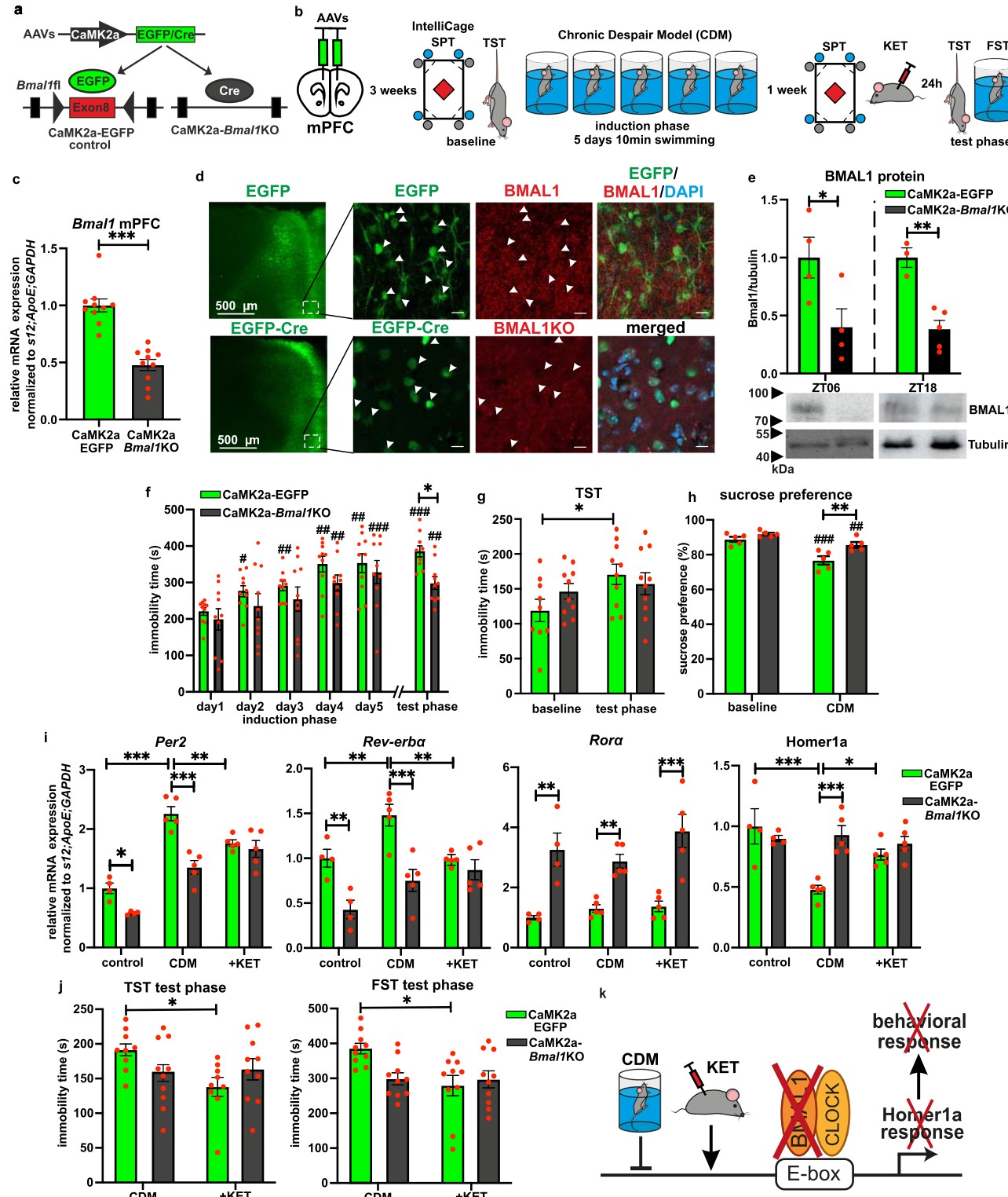

pronounced anhedonia-like behavior than controls (Fig. 5d). qRT-PCR analyses revealed SR10067-mediated decrease of *Bmal1* and Homer1a expression (Fig. 5e), while other analyzed clock genes at ZT06 under LD were not significantly affected (Supplementary Fig. 5f). Notably, acute SR10067 administered simultaneously with the rapid-acting antidepressants ketamine at ZT00 blocked its antidepressant-like effect (Fig. 5f–h).

To determine whether *Bmal1* expression in the mPFC is required for the SR10067-mediated effects, mPFC CaMK2a-*Bmal1*KO mice underwent the CDM paradigm with administration of SR10067/vehicle

after the baseline swim on day 1 (Fig. 5j). SR10067 had no significant effect on the development of immobility across the CDM induction and test phases (Fig. 5k, l), though it decreased the locomotor activity in the OFT (Supplementary Fig. 5g). In addition, expression of Homer1a was also not significantly affected (Fig. 5m). Taken together, these results suggest that the deletion of *Bmal1* in excitatory neurons of mPFC is sufficient to block the effects of SR10067 on a depressive-like phenotype.

Since potentiation of REV-ERB activity leads to an enhanced depressive-like phenotype, we then investigated the effects of

**Fig. 3 | Targeted *Bmal1* knockout in the excitatory CaMK2a neurons of mPFC affects depression-like behavior and Homer1a expression. a** Experimental strategy used for region- and cell type-specific virally induced deletion of *Bmal1*. **b** Time course of the AAV microinjections into mPFC, baseline behavioral assessment in IntelliCage followed by CDM paradigm, ketamine treatment and test phase. **c** Relative mRNA expression of *Bmal1* in mPFC at ZT06 (*n* = 10 mice per group, two-tailed Student's *t* test: ***P* < 0.001). **d** Representative images showing viral EGFP and EGFP-Cre expression in mPFC (scale bar, 500 μm) (left) and *Bmal1* labeling (red) in EGFP and EGFP-Cre expressing cells (scale bar, 20 μm) (right). **e** Representative western blot and quantitative data of BMAL1 protein expression of in mPFC at ZT06 (*n* = 4 mice per group) and ZT18 (*n* = 3 EGFP, *n* = 5 *Bmal1*KO mice), two-tailed Student's *t* test: **P* < 0.05, ***P* < 0.01). **f** Immobility time during the induction and test phase FST of *Bmal1*-floxed mice mPFC bilaterally injected with control CaMK2a-EGFP or CaMK2a-Cre AAVs. (*n* = 10 mice per group, repeated measures two-way ANOVA with Bonferroni post hoc test: **P* < 0.05 and #*P* < 0.05, ##*P* < 0.01,

###*P* < 0.001 vs day 1). **g** Immobility time during baseline and test phase TST (*n* = 10 mice per group, repeated measures two-way ANOVA with Bonferroni post hoc test: **P* < 0.05). **h** Sucrose preference assessed in IntelliCage during baseline and test phase (*n* = 5 mice, repeated measures two-way ANOVA with Bonferroni post hoc test: ***P* < 0.01 and #*P* < 0.05, ##*P* < 0.01, ###*P* < 0.001 vs baseline). **i** Relative mRNA expression of clock genes and Homer1a in mPFC at ZT06 of control (*n* = 4), CDM (*n* = 5) and CDM 24 h post ketamine mice (*n* = 5) (two-way ANOVA with Bonferroni post hoc test: **P* < 0.05, ***P* < 0.01, ****P* < 0.001). **j** Immobility time of FST and TST during the test phase of CaMK2a-EGFP/*Bmal1*KO mice 24 h post single 3 mg/kg ketamine i.p. injection (KET) (*n* = 10, two-way ANOVA with Tukey's post hoc test: **P* < 0.05). **k** *Bmal1*KO in mPFC blocks Homer1a modulation by stress and ketamine, and thus inhibits the development of depression-like behavior and the anti-depressant response. Data are presented as mean ± SEM and the individual data points are depicted. See also Supplementary Fig. 3 and Supplementary Data 1. Source data are provided as a Source Data file.

abolition of the *Rev-erbα* function utilizing a transgenic knockout mouse line (*Rev-erbα*KO)[17]. *Rev-erbα*KO mice exhibited a strong resistance to the development of a depressive-like phenotype, showing low immobility throughout the induction phase of the CDM paradigm and in the test phase FST and TST (Fig. 5n, o). Moreover, increased expression of *Bmal1* and Homer1a was observed in mPFC tissue (Fig. 5p). Negative loop genes *Per2* and *Cry1* also showed enhanced expression in *Rev-erbα*KO mice (Supplementary Fig. 5i). The *Rev-erbα*KO did not impact the locomotion of the mice, but it did produce increased anxiety-like behavior, with less time and distance spent in the center of the OFT (Supplementary Fig. 5h).

Finally, to check for off-target effects and specificity of REV-ERB agonism, SR10067 was tested on *Rev-erbα*KO mice. No significant SR10067-mediated effects were observed in immobility over the CDM induction phase, nor in the test phase FST, TST or OFT nor in *Bmal1*/*Homer1a* mRNA expression (Supplementary Fig. 5j–n).

Our data demonstrate that pharmacological activation of REV-ERB increases depression-like behavior mediated via repression of *Bmal1* and Homer1a in the mPFC (Fig. 5i).

### RORα/γ agonism elicits rapid antidepressant effects dependent on Bmal1 and Homer1a expression in mPFC

Next, we tested the effects of pharmacological activation of RORα/γ by the synthetic agonist SR1078[35,36]. WT mice acutely treated with 10 mg/kg i.p. SR1078 exhibited no behavioral difference compared to controls in locomotor activity, nor in the classical FST and TST (Supplementary Fig. 6a). Then, we investigated the effects of single SR1078 injection in CDM mice 24 h prior to the test phase (Fig. 6a). SR1078-treated CDM mice exhibited significantly increased sucrose preference (Fig. 6b) and reduced immobility in the FST and TST compared to vehicle (Fig. 6c, d), while locomotor and exploratory activity in IntelliCage was not significantly affected (Supplementary Fig. 6c). Consistently, we found increased expression of *Bmal1* and Homer1a in the mPFC (Fig. 6e) and concurrent reduction in negative loop genes *Per2* and *Cry1* (Supplementary Fig. 6b). These results demonstrate anti-depressant effects of RORα/γ agonism in the CDM model, accompanied by elevated expression of *Bmal1* and Homer1a.

To further examine the importance of the mPFC and local *Bmal1* activity, we tested the effects of SR1078 on the mPFC CaMK2a-*Bmal1*KO model. No significant difference was observed between vehicle and SR1078-treated CDM mPFC CaMK2a-*Bmal1*KO mice in the test phase FST and TST. Homer1a expression in mPFC was also not changed by SR1078 in CaMK2a-*Bmal1*KO mice (Fig. 6f–h).

Next, we investigated the necessity of Homer1a induction for the observed SR1078-mediated antidepressant effect via in vivo KD of Homer1a expression in mPFC[15] (Fig. 6i). Homer1a KD was confirmed by ~50% reduced expression in mPFC of siHomer1a injected mice compared to siCntr (Fig. 6j), while the expression of the longer Homer1b/c isoform was not significantly affected (Supplementary Fig. 6d). While

SR1078 reduced the immobility time in both FST and TST conducted during the test phase in siCntr mice, the antidepressant-like effect was blocked in siHomer1a animals (Fig. 6k). SR1078 treatment increased expression of *Bmal1* in both siCntr and siHomer1a mice (Supplementary Fig. 6e). Homer1a KD did not affect the locomotor activity in OFT (Supplementary Fig. 6f).

Finally, to control for off-target effects of SR1078, we utilized "staggerer" *Rorα*KO mice[37]. *Rorα*KO and WT CDM mice were acutely treated with vehicle/SR1078 24 h before behavioral testing. The developmentally induced complex "staggerer" phenotype made impossible the interpretation and direct comparability of the FST and OFT data to WT. Nevertheless, WT mice exhibited reduction in immobility in the TST, accompanied by an expected elevation of *Bmal1* and Homer1a expression in the mPFC, while *Rorα*KO mice showed no change in behavior or mRNA expression in response to SR1078 (Supplementary Fig. 6g).

Taken together, these results demonstrate that the Homer1a-dependent antidepressant-like effects of RORα/γ agonism in the CDM model are mediated by *Bmal1* in the mPFC.

### Pharmacological modulators of the circadian clock regulate AMPA receptors expression and homeostatic plasticity in the mPFC

In order to investigate the link between the observed antidepressant effects of circadian modulators and potential Homer1a-induced changes in the glutamatergic system[13], protein analysis was conducted on isolated synaptosomes from mPFC tissue after acute administration of REV-ERB agonist SR10067 or antidepressant RORα/γ agonist SR1078 in WT mice. SR10067 caused reduced synaptic expression of AMPA GluA1 and GluA2 receptors (Fig. 7a), while *Rev-erbα*KO mice lacking the SR10067 target receptor showed no difference from vehicle injection (Supplementary Fig. 7a). In contrast, SR1078-treated WT mice exhibited elevated expression of GluA1 at 6 h and 24 h post injection, while GluA2 was elevated at 24 h post injection only (Fig. 7b). In *Rorα*KO mice lacking the target receptor of SR1078, no elevation of GluA1 was observed (Supplementary Fig. 7b). Moreover, SR1078 had no significant effect on synaptic GluA1 expression in the mPFC of mice with KD of Homer1a expression (Supplementary Fig. 7c).

In order to further examine a potential relationship with plasticity, we examined slow wave activity (SWA, 0.5–4 Hz) during NREM sleep. SWA has been linked to regulation of synaptic strength, with the amplitude shown to reflect synaptic strength and number[8,38]. Mice underwent the CDM protocol before being implanted with electrodes to record global ECoG signal and local field potentials (LFP) from the mPFC (Fig. 7c–e and Supplementary Fig. 7d). After administration of ketamine or SR1078 at ZT06, the SWA recorded during NREM sleep in the mPFC was significantly elevated at various points over the subsequent 24 h compared to vehicle-treated animals (Fig. 7d). In contrast, significant elevation in global ECoG signal was observed only

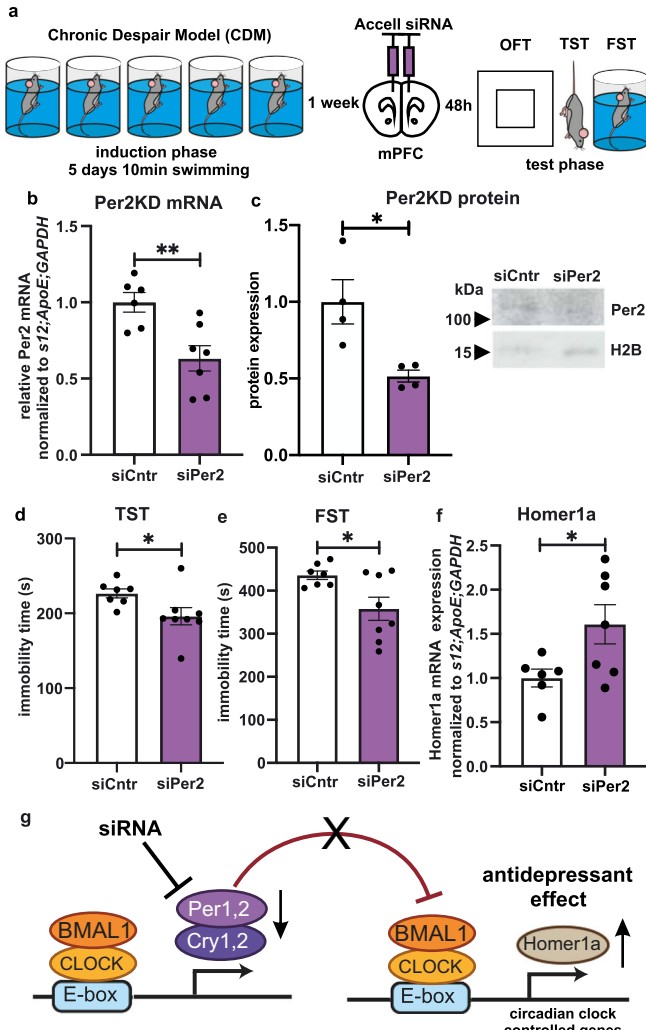

**Fig. 4 | Specific knockdown of *Per2* in mPFC elicits antidepressant effects.**
**a** Schematic overview of the experimental design: CDM paradigm, followed by stereotactic microinjections of Accell siRNA and test phase. **b** Relative mRNA expression of *Per2* in mPFC at ZT06 ($n = 6$ siCntr, $n = 7$ siPer2-injected mice, two-tailed Student's *t* test: \*\**P* < 0.01). **c** Representative western blot (right) and quantitative data (left) of Per2 protein expression normalized to Histone 2B (H2B) in mPFC at ZT06 ($n = 4$ mice per group, two-tailed Student's *t* test: \**P* < 0.05).
**d** Immunity time during test phase TST of siCntr ($n = 7$) and siPer2 ($n = 8$) injected mice, two-tailed Student's *t* test: \**P* = 0.0431). **e** Immobility time during test phase FST of siCntr ($n = 7$) and siPer2 ($n = 8$) injected mice, two-tailed Student's *t* test: \**P* = 0.0231). **f** Relative mRNA expression of Homer1a at ZT06 in mPFC of siCntr ($n = 6$) and siPer2 ($n = 7$) injected mice (two-tailed Student's *t* test: \**P* < 0.05).
**g** Schema summarizing the effects of Per2 knockdown in mPFC on Homer1a expression and depression-like behavior. Data are presented as mean ± SEM and the individual data points are depicted. See also Supplementary Fig. 4 and Supplementary Data 1. Source data are provided as a Source Data file.

immediately following ketamine administration, suggesting a stronger and/or region-specific effect in the mPFC (Fig. 7e). None of the treatments had a significant effect on the time spent in the different vigilance states (wake, SWS, REM sleep) (Supplementary Fig. 7e–g).

These results provide evidence for a correlation between the circadian modulation-derived antidepressant-like effects and known mechanisms of synaptic plasticity, evidenced by elevated AMPAR expression (Fig. 7f) and SWA during sleep.

## Discussion

Evidence for a dysregulation of circadian function in depression is accumulating, requiring precise demonstrations of relevant

mechanisms in pathophysiology and antidepressant effect[6,23,25]. Such changes in the molecular clock of depression-relevant extra-SCN oscillating brain regions are implicated in the development and treatment of depression[11,24]. Our results highlight the multi-faceted relationship between the circadian clockwork in the mPFC and depression-like behavior, demonstrating molecular clock disruption in response to stress, the importance of the mPFC clock's functional integrity for rapid antidepressant treatment, the potential for pharmacological manipulation and a plausible link to glutamate-dependent plasticity.

Chronic stress is a common and major risk factor for variety of mood disorders, including MDD[39]. The repetitive predictable swim stress used in the CDM model is designed to mimic everyday human stress, such as daily repetition of a unescapable stressful situation[28]. Here, we show a dysregulation of the molecular clock in the mouse mPFC after the CDM paradigm, characterized by increased expression of the negative loop genes, *Per2* and *Cry2*, alongside downregulation of the positive clock regulators *Rorα* and *β*. Similarly, the LH model caused an upregulation of clock suppressor genes, *Per1* and *Rev-erbα*, and decreased expression of positive clock regulators *Rorα* and *Bmal1*. The PFC is strongly implicated in mood disorders[12] and it expresses oscillating core clock genes[21,40–42] shown to be altered in post-mortem brains of depressed patients[24]. It has also been reported that the circadian clock in the mPFC is also vulnerable to dysregulation associated with depressive-like behavior induced by light-dark cycle reversal in mice[43]. Our results are coherent with other reports indicating that mood regulation requires functional clockwork in such depression-implicated regions: studies in stress models have also shown elevated nighttime *Cry* expression in the nucleus accumbens (NAc)[44], and dysregulation of a range of core clock genes including *Cry2* and *Per1/2* in the basolateral amygdala[45]. Loss of *Per2* rhythmicity has also been reported in ex vivo SCN tissue after chronic unpredictable stress (CUS)[22,40,46]. In contrast, to other animal models of depression, such as CUS or chronic mild stress, which utilize stressors directly affecting the circadian rhythm and central clock, the clock gene expression in the SCN and rhythmicity of the CDM mice were not changed. This difference may arise from distinctive nature and length of stress protocols, but it also might suggest that extra-SCN clock dysfunction be more instrumental in the development of symptoms. Thus, the CDM paradigm selectively highlights the importance of extra-SCN clockwork in key regions, like the PFC, in mediating the effects of stress on the development of depressive-like phenotypes.

The circadian clock and the stress response system are closely connected, and several core clock gene promoters, contain glucocorticoid (such as corticosterone or cortisol) responsive elements (GRE). These GRE could therefore act as mediators of stress on the peripheral brain circadian oscillators[42,47,48]. However, our data show that a CORT neither significantly alter clock gene expression in the mPFC, nor significantly change the sucrose preference.

Recent work in hippocampal CA1 and NAc has revealed that *Per2* suppression can elicit antidepressant-like effects, while whole-body, neuron- or glial-specific deletions produce a resilient-like phenotype[49,50]. Similarly, we were able to demonstrate that *Per2*KD in the mPFC produces antidepressant-like effects, further reinforcing the importance of the clock-negative regulators in depression-relevant areas.

We show that the antidepressant effects of ketamine are associated with a phase shift in the expression of numerous clock genes and a downregulation of several clock suppressors (with enhanced expression in CDM or LH). Conversely, ketamine increased the downregulated *Rorα* levels in the CDM mPFC. Ketamine effects on molecular clockwork have been previously reported[51,52]. In vitro, in neuronal cell culture, Bellet and colleagues demonstrated that ketamine directly alters CLOCK/BMAL1 promoter recruitment in a time-dependent manner and thus modulates clock gene transcription,

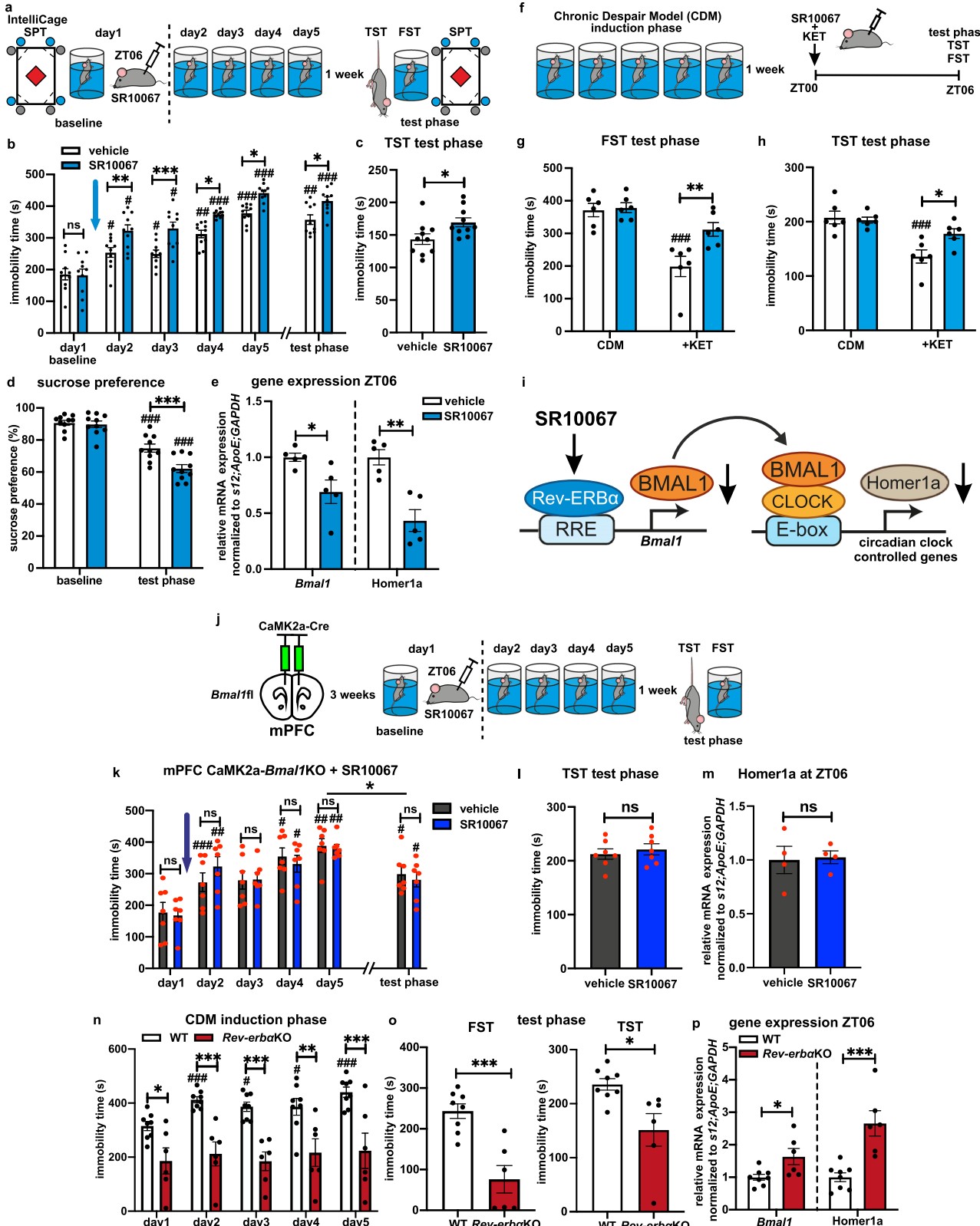

including phase-shifting of *Bmal1* and downregulation of *Per2* expression[51]. Since BMAL1 expression and activity peak during the mouse resting period, ketamine administration at ZT05 accordingly caused the downregulation of several clock genes in the anterior cingulate cortex, such as *Per1, Per2,* and *Cry2* [52]. As the half-life of ketamine is only 3–4 h, the timing of its application may play a specific role in clock modulation and thus be a factor for the magnitude and duration

of the subsequent antidepressant effect[9,10,53]. However, our data show that when applied at ZT12, ketamine still downregulates *Cry1* and *Cry2* clock repressors and increases *Rorα* expression.

Many clock gene mouse mutants have been reported to show different combinations of rhythm/mood phenotypes. A major limitation of these clock gene animal models is the lack of cell type and brain region specificity, which results in complex behavioral phenotypes[54,55].

**Fig. 5 | REV-ERBα modulates depression-like phenotype and Homer1a expression via *Bmal1*. a** Experimental design schematic. **b** Immobility time during CDM induction and test phase FST of WT mice acutely i.p. injected with vehicle/Rev-ERB agonist SR10067 (30 mg/kg) after the swim on day 1 at ZT06 ($n = 10$, repeated measures two-way ANOVA with Bonferroni post hoc test: *$P < 0.05$, **$P < 0.01$, ***$P < 0.001$). **c** Immobility time during test phase TST ($n = 10$ mice, two-tailed Student's t test: *$P = 0.0267$). **d** Sucrose preference assessed in IntelliCage ($n = 10$ mice, repeated measures ANOVA with Bonferroni post hoc test: **$P < 0.01$ and #$P < 0.05$, ###$P < 0.001$ vs baseline). **e** Relative *Bmal1* and Homer1a mRNA expression at ZT06 in mPFC of vehicle/SR10067 injected WT mice ($n = 5$, two-tailed Student's *t* test: *$P < 0.05$, **$P < 0.01$). **f** Experimental strategy: 1 week after CDM, WT mice are vehicle/SR10067 and saline/ketamine injected 6 h prior to test. **g** Immobility time during test phase FST ($n = 6$ mice, two-way ANOVA with Bonferroni post hoc test: ###$P < 0.001$ vs saline). **h** Immobility time during test phase TST ($n = 6$ mice, two-way ANOVA with Bonferroni post hoc test: ###$P < 0.001$ vs saline). **i** Schema summarizing the effects of Rev-ERB activation on *Bmal1*/Homer1a expression. **j** Experimental

design schematic. **k** Immobility time during induction and test phase of mPFC CaMK2a-*Bmal1*KO mice acutely injected with vehicle/SR10067 on day 1 at ZT06 ($n = 7$ mice, repeated measures ANOVA with Bonferroni post hoc test: #$P < 0.05$, ##$P < 0.01$, ###$P < 0.001$ vs day 1; *$P < 0.05$ day 5 vs test phase). **l** Immobility time during test phase TST ($n = 7$ mice, two-tailed Student's *t* test). **m** Relative mRNA expression of Homer1a at ZT06 in mPFC of vehicle/SR10067 injected mPFC CaMK2a-*Bmal1*KO mice ($n = 4$ mice, two-tailed Student's *t* test). **n** Immobility time during induction phase of WT ($n = 8$) and *Rev-erbα*KO mice ($n = 6$) (repeated measures two-way ANOVA with Bonferroni post hoc test: *$P < 0.05$, **$P < 0.01$, ***$P < 0.001$). **o** Immobility time during test phase FST and TST ($n = 8$ & 6 mice, two-tailed Student's *t* test: ***$P = 0.0005$ and *$P = 0.0119$). **p** Relative mRNA expression of *Bmal1* and Homer1a at ZT06 in mPFC of WT ($n = 8$) and *Rev-erbα*KO mice ($n = 6$) (two-tailed Student's *t* test: *$P < 0.05$, ***$P < 0.001$). Data are presented as mean ± SEM and the individual data points are depicted. See also Supplementary Fig. 5 and Supplementary Data 1. Source data are provided as a Source Data file.

Selective *Bmal1*KO in SCN can lead to a depressive-like phenotype, including helplessness and behavioral despair, without affecting hedonic and reward-oriented behavior[20], while specific deletion across cerebral cortical neurons produces disrupted clock gene expression profiles and increased immobility in the TST[54,56]. The brain region- and cell type-specific *Bmal1*KO mouse model used here represents a precise disruption of the circadian clockwork in CaMK2a excitatory glutamatergic neurons of the mPFC, a brain region and neuronal population strongly implicated in the pathophysiology and rapid treatment of depression[12,13,15]. In contrast to the depressive-like phenotypes reported after SCN or overall cortex KO[20,54,56], the CaMK2a-*Bmal1*KO mice exhibited both stress-resilient and antidepressant-resistant behavioral changes, suggesting that *Bmal1* expression and clock function in these particular neurons in mPFC may be an important factor in vulnerability to stress and the development and treatment of depression-like behavior[12]. As ketamine typically reduces immobility time even in naive (non-stressed) animals[52,57], an antidepressant-like effect in CaMK2a-*Bmal1*KO mice would still be expected.

An important factor in the antidepressant-like effects is the induction of Homer1a, a postsynaptic scaffolding plasticity protein which contributes to the onset of synaptic remodeling, thus impacting the efficiency of synaptic transmission[58,59] via the increased expression and activity of mGluR5 and AMPAR[13,60]. Homer1a expression is downregulated in the mPFC of animal models of stress-induced depression and its induction is necessary for the effects of several antidepressant treatments including ketamine and SD, potentially constituting a final common pathway of antidepressant mechanism[13,15,27,61]. It has been recently demonstrated that both BMAL1 and CREB bind to the *Homer1* promoter in the mouse brain[62]. Sato et al. showed that while BMAL1KO does not change the *Homer1a* baseline expression, the Homer1a immediate early response depends on BMAL1 and attenuated CREB activity[62]. Consistently, CaMK2a-*Bmal1*KO disrupted the Homer1a induction necessary for the antidepressant-like effect of ketamine[15]. Notably, the *Bmal1*KO in excitatory mPFC neurons also appeared to block the downregulation of Homer1a in response to CDM stress. BMAL1 might interact with other transcriptional regulators of Homer1a, including clock suppressors (such as *Per1, 2* and *Cry1, 2*, which are enhanced by stress, CDM and LH) or potentiators (such as GSK3β, activated by ketamine)[63,64] and thus modulate Homer1a expression and consequently the behavioral phenotype in both directions. Thus, the apparent contradiction that antidepressant-like effects are accompanied by increased *Bmal1* expression (Figs. 5m–o and 6b–e), while stress resilience is conferred by absence of *Bmal1* expression (Fig. 3f–h), is explained by the altered responsiveness of Homer1a in these conditions. Together, these data suggest that a functional mPFC clock and in particular the presence of *Bmal1* is necessary for both the development of stress-induced depression-like

behavior and the antidepressant-like effects of ketamine via blocking Homer1a downregulation or induction by stress and ketamine, respectively.

An alternative approach to influence circadian clockwork, and particularly *Bmal1* expression, is by indirect modulation of its repressor REV-ERB[17]. Manipulation of REV-ERBα has been shown to impact circadian control, with effects on midbrain dopamine activity and mood regulation[32–34]. We demonstrate here that the administration of the synthetic REV-ERB agonist SR10067[32] causes a transient anhedonic effect in DD conditions and enhances the development and maintenance of stress-induced depression-like phenotype in the CDM in LD, whilst suppressing *Bmal1* and Homer1a expression. As previously reported the distinct effects between LD and DD in response to REV-ERB agonists suggest that the light input into the SCN significantly modulates their action[34]. Moreover, the rapid antidepressant effects of ketamine were also blunted by SR10067, possibly due to downstream suppression of Homer1a. However, when *Bmal1* is absent in mPFC CaMK2a neurons, SR10067-mediated behavioral effects and downregulation of Homer1a are blocked (Fig. 5j–l).

Coherent with *Per2*KD, we also show a stress-resilient phenotype in *Rev-erbα*KO mice, accompanied by significant increases in both *Bmal1* and Homer1a. Overall, we propose that direct reduction of these negative regulatory elements, PER2 and/or REV-ERBα, lifts inhibition of BMAL1 and subsequently increases Homer1a expression, thereby producing an antidepressant-like and stress-resilient phenotype (Fig. 8). Though in many models increased depression-like behavior is associated with anxiogenic phenotype, manipulations of Rev-ERB lead to opposite effects—KO causes antidepressant and anxiogenic effect and SR10067 has pro-depressant and anxiolytic effects[32], highlighting important distinctions between circadian effects on depressive- and anxiety-related behaviors. Besides, it has been previously shown that *Rev-erbα*KO mice have enhanced neuroinflammatory response[65], which might lead to depression-like behavior. However, our experimental conditions did not represent an inflammatory challenge, which might explain the absence of an enhanced depression-like phenotype.

While SR10067 enhances the depression-like phenotype of naive mice, SR1078, an agonist of RORα/γ[35,36], exhibited antidepressant-like effects selectively in our CDM depression model with already dysregulated circadian clock in the mPFC (and no nonspecific effects in naive mice). Consistently, SR1078 also led to enhanced *Bmal1* expression and subsequent induction of Homer1a. Likewise, a strong increase in the expression of *Rorα* was detected in CaMK2a-*Bmal1*KO mice, as well as similar potentiation of *Rorα* in response to ketamine administration. *Rorα* elevation may be a result of general dysregulation of the molecular clock in the mPFC and/or compensatory mechanism in response to the *Bmal1*KO, as Bering et al. similarly reported a significant upregulation of the *Clock*-positive element in their cortical *Bmal1*KO model[56]. Crucially, we observed an inhibition of the effects of

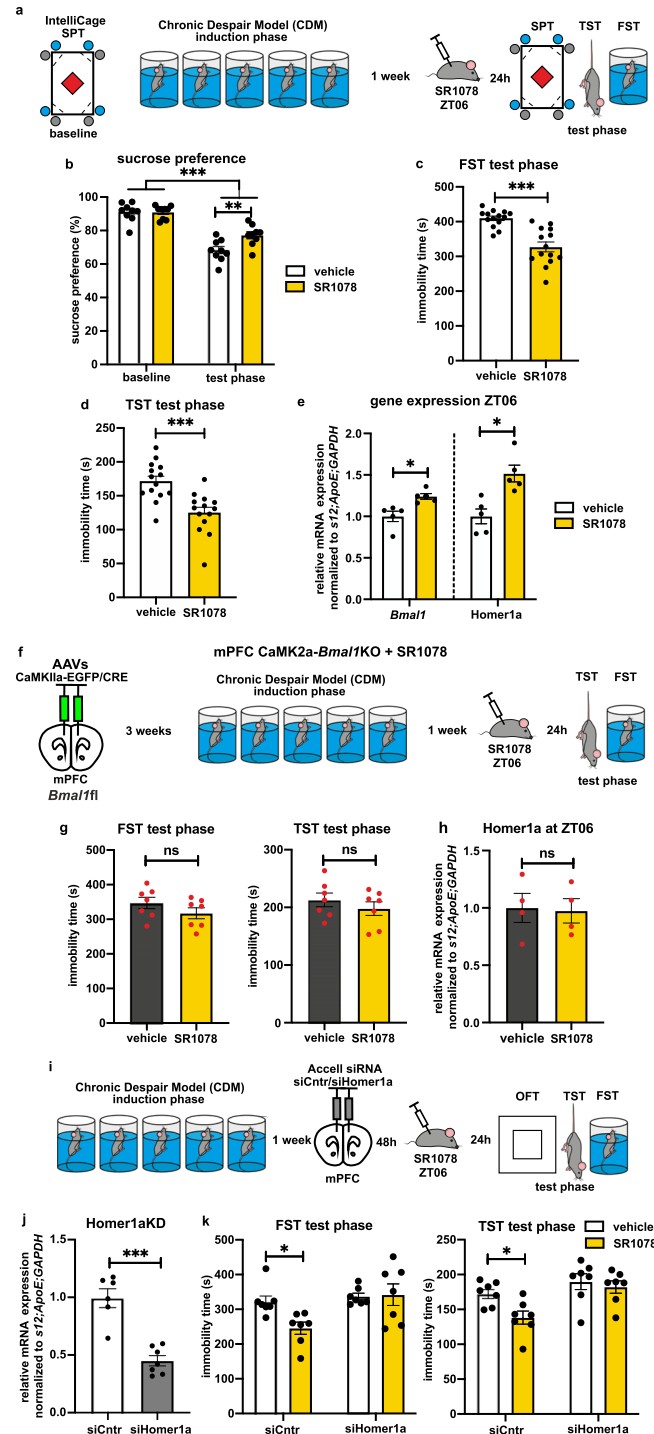

**Fig. 6 | Pharmacological activation of RORα/γ elicits rapid antidepressant effects dependent on *Bmal1* and Homer1a in mPFC. a** Experimental design: baseline behavioral assessment in IntelliCage, followed by CDM paradigm and test phase 24 h after i.p. injection of vehicle/RORα/γ agonist SR1078 (10 mg/kg). **b** Sucrose preference assessed in IntelliCage at baseline and 24 h post vehicle/ SR10067 i.p. injection (at ZT06) ($n = 9$ mice, repeated measures two-way ANOVA with Bonferroni post hoc test: **$P < 0.01$, ***$P < 0.001$). **c** Immobility time during test phase FST ($n = 14$ mice, two-tailed Student's $t$ test: ***$P < 0.0001$). **d** Immobility time during test phase TST ($n = 14$ mice, two-tailed Student's $t$ test: *$P = 0.0003$). **e** Relative mRNA expression of *Bmal1* and Homer1a at ZT06 in mPFC 24 h post vehicle/SR1078 injection ($n = 5$ mice, two-tailed Student's $t$ test: *$P < 0.05$). **f** Experimental strategy: AAV-induced mPFC CaMK2a-*Bmal1*KO, followed after 3 weeks by CDM paradigm; acute i.p. injection of vehicle/SR1078 (10 mg/kg) followed 24 h later by test phase. **g** Immobility time of test phase FST (left) and TST (right) ($n = 7$ mice, two-tailed Student's $t$ test). **h** Relative mRNA expression of Homer1a at ZT06 in mPFC 24 h post vehicle/SR1078 injection of mPFC CaMK2a-Bmal1KO mice ($n = 4$ mice, two-tailed Student's $t$ test). **i** Experimental strategy of Homer1a KD: CDM protocol on WT mice before bilateral stereotactic injection of Accell siCntr/ siHomer1a; followed by i.p. injection of vehicle/SR1078 (10 mg/kg) and test phase after 24 h. **j** Relative mRNA expression of Homer1a in mPFC of siCntr ($n = 6$) and siHomer1a ($n = 7$) injected mice (two-tailed Student's $t$ test: ***$P < 0.001$). **k** Immobility time during test phase FST (left) and TST (right) in siCntr/siHomer1a animals after administration of vehicle/SR1078 ($n = 7$ mice, two-way ANOVA with Bonferroni post hoc test: *$P < 0.05$). Data are presented as mean ± SEM and the individual data points are depicted. See also Supplementary Fig. 6 and Supplementary Data 1. Source data are provided as a Source Data file.

may also be directly influenced in part by circadian factors[66]. Although not a direct measure of AMPAR function, increased SWA may suggest greater synaptic strength and number and thus indicate enhanced plastic processes such as synaptic protein synthesis[13,15,58,59,61]. Indeed, ketamine is reported to enhance AMPAR-related glutamatergic plasticity, and to induces slow waves associated with antidepressant response[61,67,68]. In contrast, the pro-depressant effects of REV-ERB SR10067 agonism correlate with reduced SWS[32]. The interaction of the circadian clock and sleep–wake cycle has been previously implicated in the regulation of forebrain synaptic transcriptome and proteome[69,70]. Thus, considering that SWA is mechanistically related to synaptic remodeling, our results are consistent with enhanced plastic processes specific to the mPFC after SR1078 and ketamine administration, aligning with our proposed Homer1a-related mechanism. However, direct evidence of these enhanced processes is required to confirm the precise mechanisms.

Taken together, our results identify mPFC-specific clock disruption—involving amplified negative and downregulated positive regulators and misaligned *Bmal1* expression—as a key factor in the development of a depressive-like phenotype after repetitive stress. Modulation of these clock elements—via ketamine, or via direct pharmacological or genetic manipulation—contributes to the mechanism of antidepressant response, with a critical role for *Bmal1* and its control of Homer1a modulation. We also provide evidence that suppression of negative loop elements and potentiation of positive regulators may be a pathway to enhance Homer1a induction and thus increase glutamate-dependent plasticity mechanisms in the mPFC (Fig. 8). This work increases our understanding of the circadian clockwork's role in both the pathophysiology and rapid treatment of depression. Importantly, the demonstration of therapeutic modulation of the molecular clock in a model for depression provides foundational evidence for the development of novel and alternative treatments.

## Methods

### Animals

Adult wild-type C57BL/6J (RRID: IMSR_JAX:000664) and *Bmal1*-floxed mutant mice (B6.129S4(Cg)-*Arntl*^tm1Weit^/J; Jax Stock#: 007668; RRID: IMSR_JAX:007668) were sourced from Charles River (France and Germany), *Rev-erbα*KO (B6.Cg-*Nr1d1*^tm1Ven^/LazJ; Jax Stock#: 018447; RRID:

SR1078 when either *Bmal1* or Homer1a function were specifically suppressed in the mPFC, thus supporting a shared mechanism mediated by the functional mPFC clock and its local regulation of Homer1a response.

The enhanced expression of Homer1a points to synaptic remodeling via AMPAR trafficking as a potential pathway to therapeutic enhancement of plasticity[11,13,47,48,50] and related mood improvement. Supporting this hypothesis, we observed decreased synaptic AMPAR expression associated with the pro-depressive clock modulator SR10067, and increased expression associated with the antidepressant clock modulator SR1078. Administration of either ketamine or SR1078 also induced a strong elevation of SWA during sleep in the mPFC. While principally considered a marker of sleep homeostasis, SWA amplitude

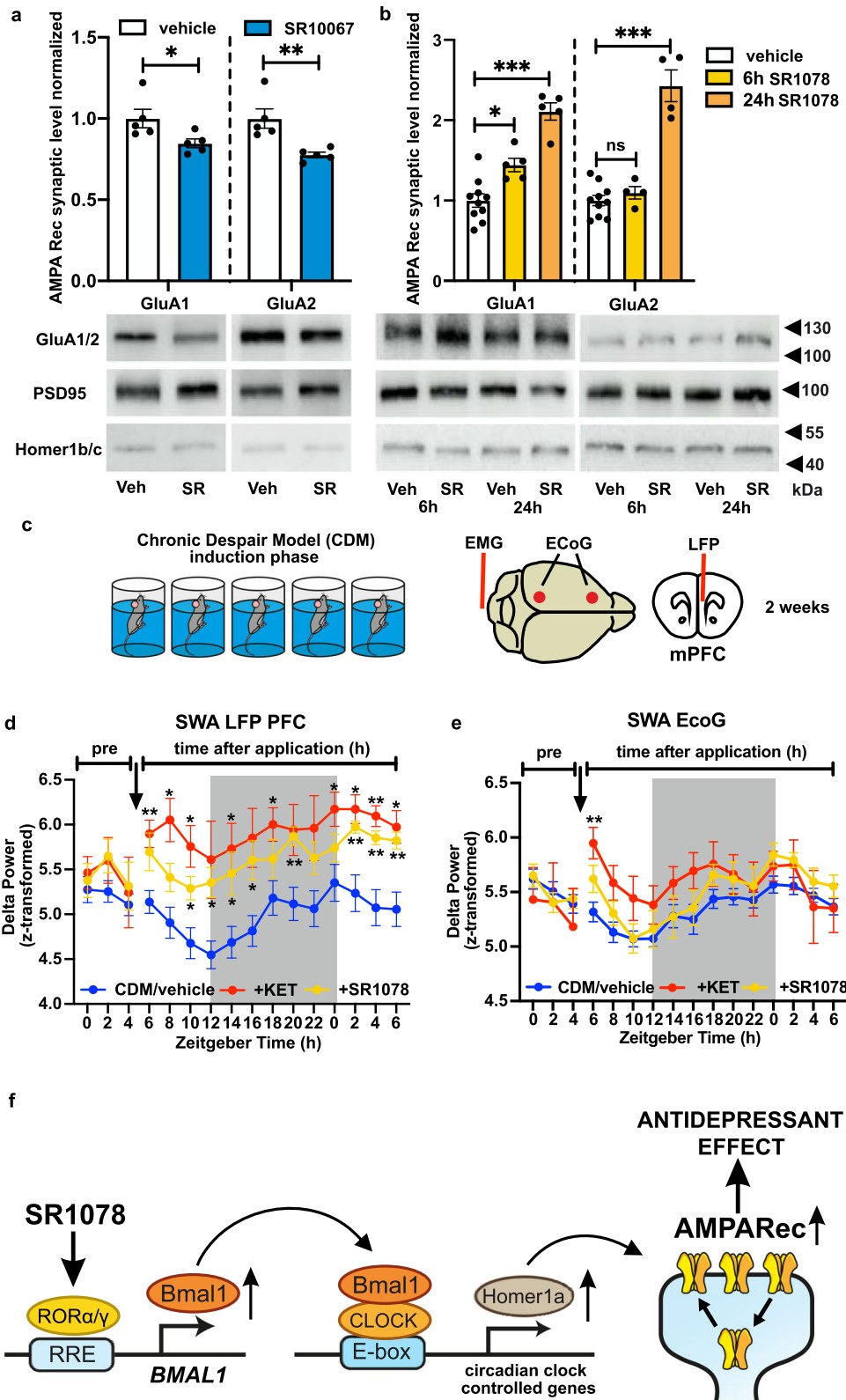

IMSR JAX:018447)[17] were sourced from the breeding pool maintained at the Chronobiotron, Strasbourg, France, and *Rora*KO staggerer mice (B6.C3(Cg)-*Rora*$^{sg}$/J; Jax Stock #:002651; RRID: IMSR JAX:002651)[37] were obtained from the breeding pool at Centre de Recherche St-Antoine (CRSA), Sorbonne Université, Paris, France. Both male and female mice were used throughout the whole study and the sex of the mice for each individual experiment is indicated in Supplementary

Table 1. All mice were at least 8 weeks old at the start of experimental procedures and housed in a temperature- and humidity-controlled environment with *ad libitum* access to food and water, maintained on a 12 h light-dark cycle, unless otherwise stated, where lights on and lights off period mean resting and active phase, respectively, for the mice. The time points of the behavioral tests coincide with the time points of the mouse sacrifice/brain tissue sampling for molecular analyses. The

**Fig. 7 | Modulation of the circadian clockwork alters AMPA receptors synaptic expression and homeostatic plasticity in mPFC. a, b** Representative western blots and quantitative data of synaptic AMPA receptors subunits GluA1 and GluA2 levels normalized to Homer1b/c and PSD95 in mPFC of WT mice 24 h after SR10067 application (**a**), and 6 h and 24 h post SR1078 injection (**b**). (*n* = 5, **a**: two-tailed Student's *t* test: *\*P* = 0.0422 for GluA1, *\*\*P* = 0.0069 for GluA2, **b**: one-way ANOVA with Bonferroni post hoc test: *\*P* < 0.05, *\*\*\*P* < 0.001). **c** Experimental design: WT mice were subjected to the CDM protocol before undergoing implantation of ECoG, LFP and EMG recording electrodes. After recovery, mice were recorded in their home cage over 32 h from ZT0, with treatment (vehicle/ketamine/SR1078)

administered at ZT06. **d, e** Delta power (0.5–4 Hz) of LFP signal recorded from the mPFC (**d**) and ECoG signal (**e**) recorded during slow wave sleep (SWS) across 12 h:12 h LD conditions (*n* = 14 mice CDM/vehicle; *n* = 6 SR1078; *n* = 4 KET; repeated measures mixed-effects model two-way ANOVA with Bonferroni post hoc test *\*P* < 0.05, *\*\*P* < 0.01 vs CDM/vehicle). Black arrow indicates time of treatment administration. **f** Schema summarizing the effects of RORα/γ activation on BMAL1, Homer1a and synaptic AMPAR expression in the mPFC and the relation to depression-like behavior. Data are presented as mean ± SEM and the individual data points are depicted. See also Supplementary Fig. 7 and Supplementary Data 1. Source data are provided as a Source Data file.

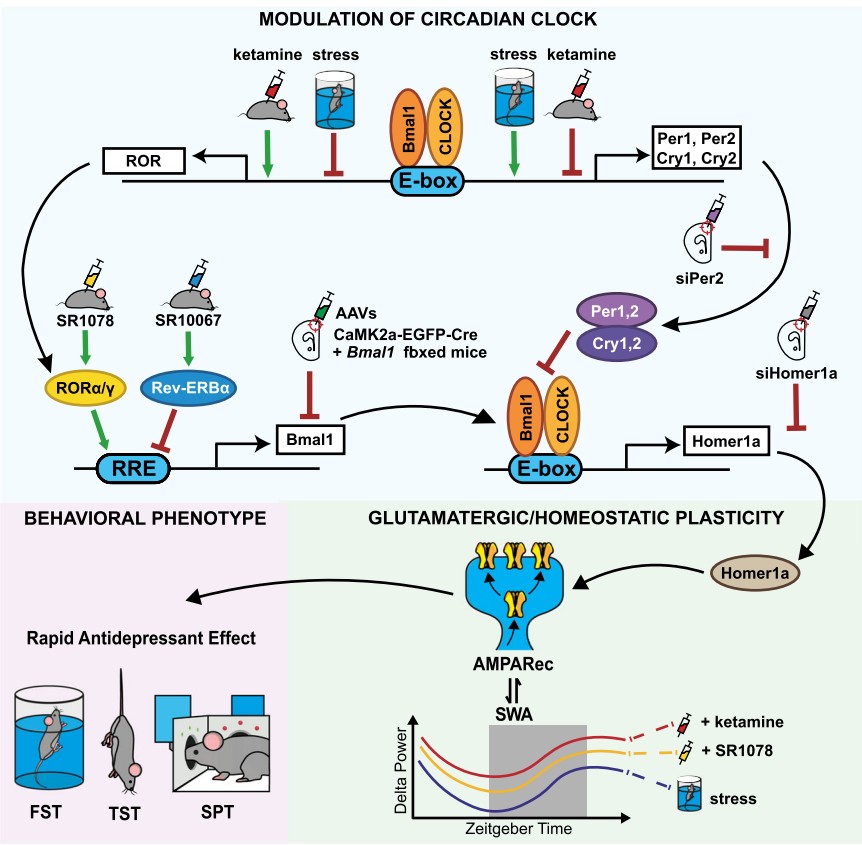

**Fig. 8 | Model of modulation of the mPFC molecular clock as a mechanism of mood-related changes.** Repetitive swim stress potentiates clock suppressor genes (*Per2* and *Cry2*) and decreases positive regulators (RORs), while ketamine opposes CDM effects. Disrupted positive and negative elements affect *Bmal1* expression and transcriptional activity, respectively, ultimately dysregulating Homer1a induction and modulating mechanisms of synaptic plasticity. Whereas, mPFC CaMK2a-*Bmal1*KO in mPFC excitatory neurons removes its regulation of Homer1a, inhibiting both pro-depressive and antidepressant responses. Experimental manipulation mimicking negative loop potentiation (Rev-ERB agonist SR10067) suppresses

*Bmal1*, Homer1a and AMPAR expression, with a pro-depressive effect. In contrast, antidepressant interventions (ketamine, RORα agonist SR1078, si*Per2* KD) inhibit negative and potentiate positive elements of the clock, increase *Bmal1* expression and activity, ultimately driving induction of Homer1a. In turn, Homer1a induction is associated with increased markers of synaptic plasticity in the mPFC including increased AMPAR expression and elevated SWA, leading to positive effects on mood. Overall, this model describes specific changes to the local mPFC clock in response to stress and rapid antidepressant treatment, highlighting *Bmal1* and Homer1a expression and subsequent plastic mechanisms as critical elements.

Chronobiotron facility is registered for animal experimentation (Agreement A67-2018-38). All procedures were performed in accordance with the German animal protection law (TierSchG), FELASA (http://www.felasa.eu), the guide for care and use of laboratory animals of the national animal welfare body GV-SOLAS http://www.gv-solas.de) and the EU Directive 2010/63/EU for animal experiments and were approved by the animal welfare committee of the Universities of Freiburg (X-20/06 R, 35-9185.81/G-19/44 and G-15/117) and Strasbourg (CREMEAS, APAFIS n°2020042818477700 and n°36945-2022042219374653v7) as well as by local authorities.

## Surgical procedures

All stereotaxic surgeries were performed under zolazepam/xylazine anesthesia (Zoletil 50 (Virbac, France), 40 mg/kg, Paxman (Virbac,

France) 10 mg/kg; i.p.). Prior to incision, animals received analgesia via subcutaneous NSAID meloxicam (Metacam (Boehringer Ingelheim, Germany), 2 mg/kg) and local anesthetics lidocaïne and bupivacaïne at the incision site (Lurocaïne (Vetoquinol, Canada)/Bupivacaïne (Pfizer, USA), 1 mg/kg). Eyes were covered with ophthalmic gel (Ocrygel, TVM, UK) to prevent drying of the cornea, and body temperature was maintained with a heated pad. Hydration was managed during and following surgery via subcutaneous saline injection. Stereotactic coordinates were taken from bregma with the skull fixed in the flat head position, with dorsal-ventral measurements taken from the level of the dura. The mPFC was targeted using the following coordinates (in mm): anterior-posterior +1.70; medio-lateral ±0.35; dorso-ventral −2.20. Following surgery, animals received atipamezole to reverse the effects of xylazine

(Revertor (Chanelle Pharma, Ireland), 0.1 mg/kg, s.c.) and further meloxicam delivered via drinking water for three days (Inflacam (Virbac, France), 5 mg/kg). With the exception of animals receiving siRNA injections, all animals were allowed a minimum 7 days recovery period before further experimental procedures.

## In vivo stereotaxic microinjections of recombinant adeno-associated viral vectors and Accell siRNA

In vivo injection of Cre- and EGFP control viruses (pENN.AAV.CaM-KII.HI.GFP-Cre.WPRE.SV40 was a gift from James M. Wilson (Addgene viral prep #105551-AAV9; http://n2t.net/addgene:105551; RRID: Addgene_105551) and pAAV-CaMKIIa-EGFP was a gift from Bryan Roth (Addgene viral prep #50469-AAV9; http://n2t.net/addgene:50469; RRID: Addgene_50469), respectively), followed the same procedure, with delivery rate set at 200 nl/min with a final volume of 650 nl/side. After viral injection, mice were allowed 3 weeks before experimental procedures to ensure adequate expression.

Accell non-targeting Pool control siRNA (5′-UGGUUUACAU-GUCGACUAA-3′; 5′-UGGUUUACAUGUUUUCUGA-3′; 5′-UGGUUUA-CAUGUUUUCCUA-3′; 5′-UGGUUUACAUGUUGUGUGA-3′); Accell Mouse Per2 siRNA-SMARTpool (5′-CCGUGGAGCAGGAAGAUAU-3′; 5′-GCAUUACCUCCGAGUAUAU-3′; 5′-GUCUGAUCCUAAGGGUAGA-3′; 5′-UCAGCGUUAUUUUAUGAUU-3′) or Accell siRNA anti-Homer1a (5′-CAGCAATCATGATTAAGTA-3′) (2 mg/ml, Horizon) were injected bilaterally into the mPFC[13,15] via a Hamilton syringe fitted with a 33-gauge needle, at a rate of 0.1 μl/min to a volume of 0.5 μl/side. The injection needle was briefly left in place and slowly withdrawn (1 mm/min) following the injection. Behavioral testing was conducted 48 h following injection of siRNA.

Virus and siRNA transfection efficacy and localization were verified by qRT-PCR and immunohistochemistry.

## Drug treatment

Mice received an intraperitoneal (i.p.) injection of racemic ketamine (3 mg/kg ±ketamine hydrochloride, Sigma-Aldrich) dissolved in saline 0.9% NaCl, SR1078 (10 mg/kg, Tocris) dissolved in a Cremophor EL (Merck)/DMSO (Sigma)/saline 0.9% (Aquapharm) 15:10:75% by volume vehicle or SR10067 (30 mg/kg, Tocris) dissolved in DMSO/saline 0.9% 15:10:75% by volume vehicle at the indicated time before behavioral testing, electrophysiological recording or sacrifice. The dosage and time of application of SR10067 and SR10078 have been selected as previously described[32,35,36]. SR1078 has not shown any unspecific/off targets/cross-reactivity effects on LXRa, LXRb, and FXR receptors, as well as SR10067 has not displayed any significant activity at any other nuclear receptor or a range of other receptors, ion channels and transporters assessed in the NIMH Psychoactive Drug Screening Program[32,35,36].

## Behavioral studies

Activity and behavior of the mice were monitored using an automatic video tracking system for recording and analysis (ANY-maze, Stoelting Co.) and an IntelliCage system (TSE Systems) unless otherwise specified. For tail suspension and forced swim tests the behavior of the animals was videorecorded (Debut Professional v8.23 video capture software, NCH Software) and manually scored by two independent experimenters, both blinded to the experimental conditions. Open-field test, tail suspension test, forced swim test and the chronic despair model were performed at ZT06 (6 h after lights on at ZT00) with both male and female mice. In order to avoid aggressive behaviors only female mice were used for the IntelliCage system experiments.

## Open-field test (OFT).
The test was performed in a square arena (50 × 50 cm) with a virtual square area of 30 × 30 cm in its center. The open field is surrounded by a 35 cm high wall made of gray PVC. Mice were placed in the center of the field and allowed to move freely.

Behavior was recorded for 10 min and total distance traveled and time spent in the central area were analyzed.

## Tail suspension test (TST).
For TST, mice were attached with their tails (1–1.5 cm from the tip of the tail) to a horizontal bar located. Each trial was conducted for 6 min and immobility time was determined. Mice observed to climb their tails (>10% of total time) were eliminated from further analysis.

## Forced swim test (FST).
For the classical FST, mice were placed in a transparent glass cylinder (15 cm diameter) filled to a height of 20 cm with water (22–25 °C). Immobility time was assessed during a 10 min swim session. Mice were considered to be immobile when they floated in an upright position and made only minimal movements to keep their head above the water.

## Chronic behavioral despair model (CDM).
To induce chronic depression-like behavior, mice were subjected to repeated swim sessions in a transparent glass cylinder (15 cm diameter) filled to a height of 20 cm with water (22–25 °C) 10 min daily for 5 consecutive days (induction phase). To avoid potential circadian disruption effects the control mice were awakened and temporarily transferred to a new cage during the swim session of the CDM mice. After a break of 1 week a final swim session was carried out (test phase). The repeated exposure to swimming leads to a long-lasting increase of the immobility time and reduction of sucrose preference during the test phase. The method has been described as a model for depressive-like behavior in mice that allows reliable prediction of antidepressant effects.

## Learned helplessness (LH).
The LH procedure included two days of induction followed by a testing day[29,68]. During induction sessions, mice were individually placed in a 2-chamber shuttle box (Panlab, Harvard apparatus) where they underwent 100 randomly spaced (10, 15, 20, or 30 s) inescapable footshocks (0.45 mA, 10 s duration) over a 50 min period. Control mice were placed in the same chamber for 50 min each day, without receiving any shock. On the test day, all mice were tested for active avoidance over 15 trials of 20 s each maximum, interspaced by a fixed 30 s time interval. During this test, mice had the possibility to escape the shock delivery by crossing into the second chamber. The latency to escape each shock was automatically recorded (Packwin 2.0 software, Bioseb) and the mean latency to escape was analyzed. If this parameter was higher than 10 s, the mouse was considered as helpless. Twenty-four hours after the test, mice were killed by cervical dislocation, brains collected, frozen in isopentane and stored at −80 °C. In all, 200 μm slices were cut in a cryostat and the infralimbic/prelimbic cortical area was collected on three consecutive slices using a 2 mm puncher.

## Chronic corticosterone protocol (CORT).
For chronic CORT exposure[30] experiments, mice were exposed to CORT in drinking water for 21 days. CORT 100 μl/ml (Sigma) was dissolved in 1% ethanol (Sigma) in the drinking water. A control group received regular drinking water without CORT.

## Behavioral analysis in the IntelliCage.
The IntelliCage system (TSE Systems) allows automatic analysis of spontaneous and exploratory behavior, activity pattern, and drinking preference of up to 16 group-housed mice implanted with radio-frequency identification (RFID) transponders. The unit consists of an open common space with 4 red housings in the center and 4 recording corners. Mice have free access to food in the middle top of the IntelliCage, while water is available at the corners behind remote-controlled guillotine doors. Each corner contains two drinking bottles and permits the visit of only one mouse at a time. The scored parameters—the number and duration of visits to any of the four corners, the nosepokes toward the doors and the licks of the bottles

−were monitored by a PC-based tracking software (IntelliCage Plus, TSE Systems). Before testing, the mice were allowed to adapt to the IntelliCage for at least 5 days with water available ad libitum in all corners.

**Sucrose preference in IntelliCage (Nosepoke paradigm).** After the initial adaptation period, the animals were habituated for 2 days to the sucrose taste: in each corner one of the bottles was filled with 1% sucrose solution and the other one with water. Both doors in the corner were open allowing free choice between the bottles. Next, a nosepoke adaptation period was carried out, where all doors were closed and the mice are required to perform a nosepoke to open them. The opened door closes automatically after 5 s of drinking. In all tasks involving sucrose-filled bottles, the positions of the bottles were exchanged every 24 h. The Nosepoke SPT protocol was used[13] for the measurement of sucrose preference with gradually increasing effort (number of nosepokes) to reach the sucrose bottles for a short period of time (12 h). In this paradigm each door opens in response to a nosepoke and closes after 5 s licking. The number of nosepokes needed to open a door to a side with a sucrose containing bottle gradually increases (1, 2, 3, 4, 5, 6, 8, 10, 12, 16, 20, 24) after every ten sucrose licking sessions. The positions of paired bottles are switched daily to reduce any confound produced by a side bias. For each bottle the number of licks was recorded and the averaged sucrose preference was calculated as percentage of the total number of licks.

## Quantitative real-time PCR (qRT-PCR)

Mice from each experimental group were killed by cervical dislocation. The brains were rapidly removed, coronally cut and the regions of interest were microdissected, quickly frozen on dry ice and stored at -80°C until used for RNA isolation. The RNA extraction and expression analyses were performed as previously described[15]. Briefly, the tissues were homogenized in guanidine thiocyanate/ 2-mercaptoethanol buffer and total RNA was extracted with the sodium acetate/phenol/chloroform/isoamylalcohol procedure. Then, samples were isopropanol-precipitated and washed twice with 70% ethanol. The pellets were dissolved in RNase-free Tris-HCl buffer (pH 7.0) and RNA concentrations were determined with a spectrophotometer (Bio-Photometer; Eppendorf). Reverse transcription was performed with 1 mg of total RNA using M-MLV reverse transcriptase (Promega). Quantitative real-time PCR was performed on a 7300 Real-Time PCR System with Sequence Detection Software qPCR v1.3.1 (Applied Biosystems) using Fast SYBR™ Green Master Mix (4385612, Applied Biosystems). All qRT-PCR experiments were performed blinded as the coded cDNA samples were pipetted by a technician. The target gene mRNA levels were normalized to the levels of apolipoprotein E (*ApoE*), glyceraldehydes-3-phosphate dehydrogenase (*GAPDH*), and *s12* RNA. The primer sequences used were as follows− *ApoE*: 5′-CCTGAA CCGCTTCTGGGATT-3′, 5′-GCTCTTCCTGGACCTGGTCA-3′; *GAPDH*: 5′-TGTCCGTCGTGGATCTGAC-3′, 5′-CCTGCTTCACCACCTTCTTG-3′; *s12*: 5′-GCCCTCATCCACGATGGCCT-3′, 5′-ACAGATGGGCTTGGCGCT TGT-3′; *Per1*: 5′-CCAGATTGGTGGAGGTTACTGAGT-3′, 5′-GCGAGAGT CTTCTTGGAGCAGTAG-3′; *Per2*: 5′-AGAACGCGGATATGTTTGCTG-3′, 5′-ATCTAAGCCGCTGCACACACT-3′; *Cry1*: 5′-AGGAGGACAGATCCCA ATGGA-3′, 5′-GCAACCTTCTGGATGCCTTCT-3′; *Cry2*: 5′- GCTGGAAG-CAGCCGAGGAACC-3′, 5′-GGGCTTTGCTCACGGAGCGA-3′; *Bmal1*: 5′-CTCCAGGAGGCAAGAAGATTC-3′, 5′-ATAGTCCAGTGGAAGGAATG-3′; *Clock*: 5′-GGCGTTGTTGATTGGACTAGG-3′, 5′-GAATGGAGTCTC-CAACACCCA-3′; *Npas2*: 5′-ACGCAGATGTTCGAGTGGAAA-3′, 5′-CGCC CATGTCAAGTGCATT-3′; *Rev-erbα*: 5′-CCCTGGACTCCAATAACAA-CACA-3′, 5′-GCCATTGGAGCTGTCACTGTAG-3′; *Rorα*: 5′-TTGCCAAAC GCATTGATGG-3′, 5′-TTCTGAGAGTCAAAGGCACGG-3′; *Rorβ*: 5′-ATGG CAGACCCACACCTACG-3′, 5′-TATCCGCTTGGCGAACTCC-3′; *Rory*: 5′-CGAGATGCTGTCAAGTTTGGC-3′, 5′-TGTAAGTGTGTCTGCTCCGC G-3′; Homer1a: 5′-CAAACACTGTTTATGGACTG-3′, 5′-TGCTGAATT-GAATGTGTACC-3′.

## Preparation of nuclear and synaptosomal extracts

Mice were killed by cervical dislocation and the dissected brain regions were mechanically homogenized in homogenization buffer (320 mM sucrose, 4 mM HEPES [pH 7.4], 2 mM EDTA) and centrifuged at $800\times g$ for 15 min at 4 °C to generate total (S1) and nuclear fraction (P1). All buffers contained a phosphatase (P5726-1ML) and protease inhibitor (P8340-1ML) cocktail (Sigma-Aldrich). Synaptic (P2) and cytosolic (S2) fractions were obtained by centrifugation of S1 at $10,000\times g$ for 15 min at 4 °C. Washed synaptosomal pellets (P2) were directly lysed in lysis buffer (50 mM Tris-HCl [pH 6.8], 1.3% SDS, 6.5%, glycerol, 100 mM sodium orthovanadate) containing phosphatase and protease inhibitor cocktail (Sigma-Aldrich) and boiled for 5 min at 95 °C and processed for SDS-PAGE and western blot. Nuclear fraction was homogenized in buffer containing 50 mM Tris/HCl (pH 7.0), 10 mM EDTA, 1 mM sodium orthovanadate, 0.01%Triton X-100 and proteinase inhibitor cocktail, shortly sonicated, centrifuged at $1000\times g$ and then the supernatant was mixed with lysis buffer and further processed as described above. The protein concentration of the different fractions was determined using a BCA assay kit (Thermo Fisher Scientific) according to the manufacturer's instructions.

## Western blot

After adding 10 mM DTT, synaptosomal (P2) lysates were boiled for 5 min at 95 °C, separated by SDS-PAGE on 7.5% acrylamide gels and transferred to nitrocellulose transfer membrane (Whatman). The membranes were blocked with 5% non-fat dry milk in TBS-T (1% Tween20 in Tris-buffered saline (TBS)) and afterward incubated with the respective primary antibodies diluted in TBS (mouse anti-GluR1-NT, Merck (MAB2263), 1:1000; rabbit anti-GluR2, Merck (AB1768-I), 1:1000; rabbit anti-Homer1, Merck (ABN37), 1:1000; mouse anti-PSD95, Merck (MABN68), 1:1000; rabbit anti-mouse Per2, Alpha Diagnostics (PER21-A), 1:1000; rabbit anti-BMAL1, Thermo Fisher Scientific PA1-523, 1:1000; mouse anti-Histone H2b, Euromedex (IG-H2-2A8), 1:500 overnight at 4 °C. After 3 times washing membranes were incubated with horseradish peroxidase-conjugated secondary antibodies diluted in TBS-T (sheep anti-mouse, GE Healthcare (NA931), 1:20,000; donkey anti-rabbit, GE Healthcare (NA9340), 1:25,000) for 1 h at room temperature. Washed membranes were developed with an Amersham Imager 680 (GE Healthcare) using an enhanced chemiluminescence detection kit (Thermo Fisher Scientific). Band intensity was quantified by densitometry with ImageJ 1.53 m software (National Institute of Health, USA) and normalized to the appropriate loading control.

## Immunohistochemistry

Animals were anesthetized with a mix of Ketamin-Rompun (Ketamine [CP Pharma] 50 mg and Rompun [Bayer Healthcare] 0.5 mg per 100 g body weight) and transcardially perfused with 50 mL of ice-cold PBS (8.1 mM $Na_2HPO_4$, 138 mM NaCl, 2.7 mM KCl and 1.47 mM $KH_2PO_4$ [pH 7.4]). Brains were removed, postfixed overnight in 4% paraformaldehyde in PBS at 4 °C, and cryoprotected for 2 days in 30% sucrose in PBS at 4 °C. The brains were then frozen, and 40-µm coronal sections were cut with a sliding cryostat (Leica Microsystems). Then, the free-floating sections were incubated with blocking solution (0.3 % Triton X-100 and 5% normal goat serum in PBS) for 1 h at 4 °C. Sections were incubated with the respective primary antibody− mouse anti-GFAP, Cy3 Conjugate (1:1000; Merck MAB3402C3), mouse anti-CaMK2a (1:1000; Invitrogen Cba-2 13-7300) and rabbit anti-BMAL1 (1:1000; Thermo Fisher Scientific PA1-523) in blocking solution at 4 °C overnight. After 3× washing with PBS, sections were incubated with the secondary antibody− Alexa Fluor 594 goat anti-mouse (1:1000; Thermo Fisher Scientific A-11032) and goat Anti-rabbit IgG H&L (Cy5 ®) (1:1000; Abcam ab6564) in blocking solution for 3 h at room temperature. Sections were then washed and stained with 1 mg/ml 4′, 6-diamidino-2-phenylindole (DAPI, 1:1000; Thermo Fisher Scientific 62248) for 10 min. After the final washes in PBS, slices were mounted on slides

using mounting medium from Life Technologies (E6604). All immunofluorescence images were detected and photographed with an AXIO Imager.M2 fluorescence microscope and analyzed using ZEN 3.5 software (Carl Zeiss) and ImageJ 1.53 m (National Institute of Health, USA).

## In vivo electrophysiology recordings

Two electrocorticogram (ECoG) electrodes (constructed from 1.2 mm gold-plated steel screws and 60 μm-diameter Teflon-coated tungsten wire (World Precision Instruments, USA)), inserted to the level of the dura over the frontal and parietal regions, and a pair of monopolar local field potential (LFP) electrodes (60 μm-diameter Teflon-coated tungsten wire (World Precision Instruments, USA)) targeting the mPFC were inserted during stereotaxic surgery (as described above). Additional screw electrodes were inserted to act as reference and ground, and a further electrode (60 μm-diameter Teflon-coated tungsten wire (World Precision Instruments, USA)) was inserted into the nuchal muscle to record electromyogram (EMG). Electrodes were fixed to the skull with Superbond (Sun Medical), connected to a connecting headpiece (P1 Technologies) and fixed in place with dental cement (company). After recovery from surgery, animals were attached via flexible cable to a rotating joint (P1 Technologies), allowing free movement in the home cage. Animals were habituated to the recording set up for 72 h before any recording procedure. During recording sessions, electrophysiological signals were amplified, digitalized and sampled at 500 Hz (Neuvo 64-channel amplifier and ProFusion software package, Compumedics, Australia), and data were stored for offline analysis.

**Sleep scoring.** Analysis of vigilance states was performed manually using ProFusion software (Compumedics, Australia). ECoG and EMG signals were divided into 4 s epochs and each 4-s epoch was classified according to standard criteria as wake (identified by ECoG activity in the theta (8–12 Hz) and high-frequency ranges, with concurrent EMG activity), SWS (identified by characteristic slow waves of low-frequency, high-amplitude ECoG activity and low EMG) or REM sleep (identified by characteristic theta and minimal EMG activity). After scoring, sleep data parameters were averaged over periods of 1 h.

**Spectral analysis.** ECoG and LFP signals were imported into MATLAB (Mathworks, USA) for spectral analysis using the Chronux and MATLAB Signal Processing (Mathworks, USA) toolboxes. Line noise was removed using a 512-point Hanning window, and values exceeding 3 standard deviations of the raw data identified as artefacts and removed with 1 s adjacent data. The signal was separated according to vigilance state (see above), concatenated, bandpass filtered between 0.5 and 200 Hz and transformed using a multi-taper method (time-bandwidth product of 3, using 5 Slepian tapers with a 2 s window moving at 0.1 s) implemented via the Chronux toolbox. Data were z-transformed and averaged over 1 h periods prior to statistical analysis.

## Statistical analysis

All values are expressed as means ± SEM. Statistical analyses were performed with GraphPad Prism 8.0.0 software (GraphPad Software) using one- or two-way analysis of variance (ANOVA) followed by Bonferroni's or Tukey's post hoc tests to compare the means of two or more groups or unpaired two-tailed Student's t test to compare the means of two groups. The daily rhythms of clock gene expression were evaluated by cosinor analysis. Sine waves (least-squares regression) with the frequency constrained to exactly 24 h were then fitted to each of these graphs in order to better visualize time of peak and trough for each region and gene using the following equation: $y = A + B*\cos(2\pi [x − C]/24)$ where A is the mean level, B is the amplitude, and C is the acrophase of the fitted rhythm. To further compare the amplitude and the acrophase of the curve fits generated from the sine wave model of control, CDM and ketamine the extra sum of squares F test was performed. A P value ≤ 0.05 was considered to be significant (*$P ≤ 0.05$, **$P ≤ 0.01$, ***$P ≤ 0.001$). Prior to statistical analyses data assumptions (for example normality and homoscedasticity of the distributions) were verified using D'Agostino–Pearson, Shapiro–Wilk, and Kolmogorov–Smirnov tests (GraphPad Prism 8.0.0 software). Detailed statistical approaches and results are provided in Supplementary Data 1. Statistical analysis summaries are mentioned in the figure legends. For all molecular and behavioral studies mice were randomly assigned to the groups. Most behavioral and molecular results were confirmed via independent replication of the experiments as shown in Supplementary Data 1. In addition, investigators were blinded to the treatment group until data have been collected. Sample sizes were determined on the basis of extensive laboratory experience and were verified via power analysis.

## Reporting summary

Further information on research design is available in the Nature Portfolio Reporting Summary linked to this article.

## Data availability

The datasets generated and/or analyzed during the current study can be found in the paper and the supplementary materials. Any additional research materials will be made available on a request at serchov@inci-cnrs.unistra.fr. Source data are provided with this paper.

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

## Acknowledgements

The authors thank Dr. James M. Wilson and Dr. Bryan Roth for distributing the pENN.AAV.CamKII.HI.GFP-Cre.WPRE.SV40 and pAAV-CaMKIIa-EGFP viruses, Dr. Ueli Schibler (University of Geneva, Switzerland) for providing the founder *Rev-erbα*KO mouse line, Anne-Laurence Boutillier for offering anti-Histone H2b antibody, Aurelia Ces and the CompOpt core facility of the UPR3212 for the technical support, Dr. Dominique Ciocca and Dr. Sophie Reibel as well as the whole core facility Chronobiotron UMS3415 (Strasbourg, France) for animal care and animal ethical experimentation support. The study was funded by grants from the German Research Council (SE 2666/2-1 and SE 2666/2-3) to T.S., Medical Research Foundation (FRM) (AJE201912009450 to T.S. and FDT202304016976 to M.S.), Sigrid Juselius Foundation to T.R., University of Strasbourg Institute for Advanced Study (USIAS) (2020-035) to T.S., University of Strasbourg ITI NeuroStra (as part of the ITI 2021-2028 program of the University of Strasbourg, CNRS and Inserm, supported by IdEx Unistra ANR-10-IDEX-0002 under the framework of the French Program *Investments for the Future*) to T.S., the French National Research Agency (ANR) through the Programme d'Investissement d'Avenir EURIDOL graduate school of pain ANR-17-EURE-0022 to M.S.; Centre National de la Recherche Scientifique (CNRS UPR3212); the Région Grand-Est (Fonds Régional de Coopération pour la Recherche, CLueDol project) for CompOpt equipment. B.A. receives support as a CNRS-affiliated researcher.

## Author contributions

Conceptualization: T.S., P.B. and E.C.; methodology: D.S., W.G., C.M., M.B., M.V. and T.S.; investigation: D.S., W.G., T.S., C.M., C.R., S.R., M.V. and S.V.; resources: P.B., E.C., T.R., B.A. and C.N.; writing—original draft: W.G., D.S. and T.S.; writing— review and editing: W.G., T.S., D.S., M.S., M.B., E.C., T.R., P.B., S.V., M.V. and C.N.; funding acquisition: T.S.; supervision, T.S., P.B., E.C., M.B. and C.N.

## Funding

## Competing interests

T.S. is an honoraria consulting and advisory board member of Primetime Life Sciences, LLC. C.N. received lecture fees and advisory board honoraria from Janssen-Cilag. The remaining authors declare no competing interests.
