## [Peer Review File · Nature Communications]

Prefrontal cortex molecular clock modulates development of depression-like phenotype and rapid antidepressant response in miceREVIEWER COMMENTS

Reviewer #1 (Remarks to the Author):

This is an intriguing piece of work that could significantly contribute to our understanding of the link between the circadian clock and rapid antidepressants. The authors argue that BMAL within the CAMK2a neurons of the mPFC is necessary to induce the ketamine-mediated antidepressant effect through the direct induction of the transcription factor HOMER1a, which, in turn, targets plasticity-related genes. The authors propose a mechanism involving the transcriptional regulator RORa, a known BMAL inducer. Along these lines, the authors argue that the inhibition of BMAL1 by its classic transcriptional repressors PER2, CRY2, or REVERBa induces depressive behavior. The experiments are well-designed, and the authors have employed various complex techniques. Therefore, I believe it is worth considering for publication in Nature Communications after major revisions and addressing the concerns raised in this review.

The manuscript has three main and critical weaknesses that need to be appropriately addressed.

Since most of the work is based on the assumption of BMAL's role in the antidepressant effects of different drugs, the elimination within the CAMK2a neurons is not convincing (Fig. 2a, b, and Supplementary Fig. 2a). In Figure 1b, the authors should demonstrate the expression of BMAL1 with or without CRE, which would reflect the result in panel C. The merged figure should clearly illustrate a reduction in the colocalization between BMAL1 and CAMK2A in the mice expressing CRE. Additionally, corresponding quantification should be included.

The text is challenging to follow; the language should be clearer, more concise, and precise to be convincing. The discussion section is extensive, somewhat redundant, and requires a better bibliographical analysis to support the model. Therefore, the manuscript would benefit from careful revision.

On this note, the model appears incomplete, lacks sufficient support, and presents some results that seem contradictory due to the absence of thorough analysis, especially concerning the role of BMAL1. For instance, the authors could explore the involvement of other transcription factors contributing to the regulation of BMAL or RORa, which, under certain circumstances (e.g., stress, ketamine, sleep deprivation), may have either antidepressant or depressive effects. Furthermore, integrating acrophase data from different genes into the results and/or the model would provide valuable insights. Additionally, it would be beneficial if the authors discussed the distinctions between depression-like behavior and anxiety, as they observed different effects on these behaviors.

Specific points

1. "We found increased oscillation of clock negative regulatory loop" It is somewhat ambiguous. Is the author referring to the transcription of clock genes or the negative regulators like PER or CRY?
2. "normalized by rapid antidepressant ketamine" What does it mean to 'normalized' the increase in the oscillation?
3. "clockwork via viral Bmal1KO in CaMK2a excitatory neurons blocks both development of depressive-like phenotype and ketamine's effect." This could be explained more thoroughly because ketamine is an antidepressant, unlike stress-induced depression.
4. "Repetitive swim stress and rapid antidepressant ketamine alter circadian clock gene expression in the mouse mPFC" These are opposing stimuli; therefore, the title is somewhat confusing.

5. "mice (1 week after the induction phase)" What does 'after the induction phase' mean?
6. In Table 1b of the supplementary materials, where only P values are provided, it is not clear how the values change. For instance, in the rhythmicity P values, it would be helpful to represent them as $-\log P$ values and include a color or bar scale. For comparative values, consider using a color scale to show the direction of the change, in addition to the $-\log P$ value.
7. "(except Per1)" Except Per1, In the supplementary table, the Per1 values are similar to those of the other genes; therefore, this statement is not clear
8. In Supplementary Table 1E, it is not clear what the P-value of the amplitude is referring to, nor the CDM effect. Could you please specify which values are being compared?
9. "including amplification of negative loop" This is not clear. What do the authors mean by "amplification"? Are they referring to increased amplitude or more robust rhythms?
10. "normalization of Bmal1 oscillation" This is somewhat ambiguous. Does this normalization refer to a phase shift in the acrophase?
11. "potentiation of positive element Rora" The same applies to RORA. It would be better to clarify whether there's an increase in amplitude or a strengthening of the rhythms.
12. "Targeted Bmal1 knockout in the excitatory CaMK2a neurons of mPFC blocks Homer1a modulation by stress and ketamine and inhibits the development of depression-like behavior and the antidepressant response. " The title is confusing. It should separate the ketamine and stress responses, as these are antagonistic and do not sound biologically coherent.
13. Supplemental Figure 2b should include the word 'baseline' in the figure title
14. mice (Fig. 2d) " Figure 2d refers to TST or FST?
15. "thus preventing negative clock element upregulation by CDM (Fig. 2i and Supplementary Fig. 2h)." This statement is confusing. What does 'negative clock element' mean? Additionally, as RORA is not a repressor, it should be discussed in a separate sentence.
16. "and decreased the expression of Per1, Per2 and Rev-erba " rather than decreasing to prevent the CDM-mediated induction of the expression of..."
17. "However, the Bmal1 deletion in mPFC CaMK2a neurons inhibited the antidepressant effects of KET (Fig. 2g, h). Or the elimination of BMAL1 induces antidepressant effects.
18. Fig. 2k This model is confusing. Perhaps it could be split into two parts: one referring to BMAL1 and the other to KET, both of which are inducers of Homer1 and, consequently, associated with antidepressant effects, rather than solely focusing on the effect of eliminating BMAL. Additionally, when mentioning 'Behavioral response,' please specify the type of behavior, whether it pertains to depressive or non-depressive behavior.
19. "transient hypoactive agent" What does 'hypoactive agent' mean? This term is confusing. It would be better to describe the specific effect induced by this agonist.
20. "acute or chronic 5 days injection of SR10067" It is unclear whether 'DD' corresponds to acute or 5-day SR10057 injections.
21. "induced anhedonia-like behavior 48h after injection and phase shift of the locomotor activity (Supplementary Fig. 4c, d). " In Supplementary Figure 4A, the schema shows that SR10067 was tested under both LD and DD conditions. However, in Supplementary Figure 4C, there is only one actogram of SR10067 under DD. To enhance clarity and persuasiveness, it would be necessary to include two actograms (one for the vehicle and one for SR10067) covering both LD and DD periods and depicting the days of SR10067 injections.
22. "the effects of single" It appears that DD itself induces some anhedonia, as reflected at 74 hours, while at this time, SR no longer has any effects. How can this be explained?
23. In Figure 4B, is it the Forced Swim Test (FST) or the Tail Suspension Test (TST)?
24. "Throughout CDM induction, SR10067- treated mice exhibit consistently elevated immobility time compared to vehicle-treated controls, an effect maintained in the test phase

FST and TST (Fig. 4b, c).” In the sucrose preference test under DD, the anhedonic effects of SR10067 were lost after 72 hours. However, in the TST (or FST), the depression-like effects last several days. How can this apparent discrepancy be explained?

25. Why is it that in Supplementary Figure D, there is no effect on anhedonia under LD, while in Figure 4D, there is an effect several days after SR10067 injection?

26. "To test the role of mPFC Bmal1 expression in SR10067 action" This sentence is confusing. Something more specific and explicit would be more appropriate. For example, 'To determine whether BMAL1 in the MPFC is required for the induction of SR10067-mediated depression...'

27. "(Fig. 4j).” Since the figure depicting the elimination of BMAL1 in the CaMKIIa neurons (the primary strategy) is already shown in Figure 2a, it is not necessary to repeat the same figure. Otherwise, readers might initially assume it's a different genetic experiment. Instead, the genetic strategy in Figure 2A can be isolated in one panel, while the Intellicage and CDM, KET, etc., can be presented in Panel 2B. Therefore, in Figure 4J, simply indicating the mouse genotype is sufficient.

28. "while its hypoactive and anxiolytic action were preserved" Please display the same plots for the WT mice treated with SR10067.

29. "In addition, expression of Homer1a " To provide more evidence (e.g., ChIP assays, eBOX experiments, etc.) regarding the involvement of BMAL in the regulation of HOMER

30. "these results suggest that the KO of Bmal1 expression in excitatory neurons of mPFC is sufficient to block the pro-depressive effects of SR10067. " This conclusion would be better supported by taking a more biological perspective, such as focusing the conclusion on the roles of REVERB and BMAL molecules.

31. "Supplementary Fig. 5e" The referenced figure is not correct; please correct it.

32. "(Supplementary Fig. 5d). " This figure can be placed at the end of Supplementary Figure 5.

33. "After administration of KET or SR1078 at ZT06, the SWA recorded during NREM sleep in the mPFC was significantly elevated at various points over the subsequent 24h compared to vehicle-treated animals " It could be interesting to determine circadian parameters related to SWA.

34. Since it has been demonstrated that KET shows long-term effects that last up to 15 days, it would be interesting to determine if the agonist SR1079 also exhibits long-term antidepressant effects.

35. "The brain region- and cell type- specific Bmal1KO mouse model used in our study represents a very precise disruption of the circadian clockwork as it is restricted to the CaMK2a excitatory glutamatergic neurons of the mPFC, " Is CaMK2a exclusively expressed in the MPFC? Please provide more convincing evidence or discuss this point

36. "Notably, it has been recently demonstrated that Homer1a undergoes bimodal regulation by BMAL1" Please provide more specific details

37. It is necessary to measure the expression of CREB because the increase in PERs-CRYs and the reduction of HOMER1 cannot be explained solely by the presence of E-box elements.

38. "Moreover, Homer1a expression appears inversely correlated with that of negative clock elements. " Where is it observed? Or under what conditions?

39. "Importantly, our Bmal1KO in excitatory mPFC neurons appeared to block the downregulation of Homer1a in response to CDM stress." Please refer to the corresponding figure

40. "consistent with the aforementioned inverse correlation with Homer1a," Please refer to the corresponding figure

41. "The CaMK2a-Bmal1KO also led to decreased expression of negative loop genes, consistent with the aforementioned inverse correlation with Homer1a , and potentially

contributing to stress resilience by preventing the amplification of negative clock elements typically observed after CDM ." This sentence is confusing because it does not mention a link or mechanism between negative loop genes and Homer1a expression.

42. "Simultaneously, a strong increase in the expression of the positive regulatory element Rora was detected in CaMK2a-Bmal1KO mice;" Based on the literature, the author should speculate about a common mechanism for the transcriptional regulation of Homer1a and RORa.

43. "Rora elevation may be a compensatory mechanism in response to the Bmal1KO, as Bering et al. similarly observed significant upregulation of the Clock positive element in their cortical Bmal1KO model" It would be necessary to provide a more detailed mechanism for this assumption

44. "We show that the antidepressant effects of KET administration are associated with a striking normalization of the negative loop potentiation observed after CDM. " The comparison between the amplitudes of these genes in the control, CDM, and ketamine groups should be conducted using their relative values and P-values

45. " In our CaMK2a-Bmal1KO mice, KET had neither an antidepressant effect on behavior, nor did it downregulate negative loop amplitude or upregulate Homer1a expression. " This sentence is confusing. It contains three ideas that should be separated for better understanding.

46. "Despite some inhibition of the response to stress, " What do you mean by 'response to stress'? Are you referring to a depressive effect?

47. "as KET typically reduces immobility time even in naive (non-stressed) animals, an antidepressant-like effect would still be expected. " This sentence is confusing. Can you please clarify or provide more details? an antidepressant-like effect would still be expected. "

48. "modulation of depressive-like behavior" This is ambiguous

49. "bi-directional manner via its regulation of Homer1a responsiveness to CDM (downregulation) or KET (upregulation). " Why do the authors make this statement?

50. "Further, our data show Homer1a induction is again concurrent with clock-mediated antidepressant effect." Please refer to the figure(s) or provide a hypothesis.

51. "Coherent with the Rev-erba downregulation seen after both Bmal1KO and Per2KD, we also show a stress-resilient phenotype in Rev-erbaKO mice." Given that Per2 is part of the negative feedback loop of ebox-targeted promoters, how can we explain the downregulation of REVERBa in the Per2KD?

52. "Overall, we propose that direct reduction of these negative regulatory elements, PER2 " Elements can be confused with DNA-binding motifs within promoters; therefore, I suggest referring to them by their actual names, such as transcription factors or transcriptional regulators.

53. "This interruption of effect is specific to the mPFC: although REV-ERB activity is linked to other mechanisms implicated in depression, including neurogenesis in the hippocampus, dopaminergic system and stress-responsive glucocorticoid systems⁶⁰⁻⁶², the blunted effect in our CaMK2a-Bmal1KO mice confirms again the specific importance of Bmal1 function in excitatory neurons for mood modulation." This sentence is confusing. Initially, the author highlights the importance of REVERB in depression and brain processes such as neurogenesis. However, the text then shifts to discussing BMAL without establishing a clear link between these two proteins.

54. "Similarly, lack of SR10067 effect in Rev-erbaKO mice confirms that the pro-depressive effects are not due to other off-target interactions of SR10067." Perhaps something like the absence of SR10067-induced pro-depressive effects would be more appropriate...

55. The induction of Homer1a is the critical driver in relation to mood; the apparent contradiction that antidepressant-like effects are accompanied by increased Bmal1 expression, while stress resilience is conferred by absence of Bmal1 expression" Please

refer to figures.

56. "The induction of Homer1a is the critical driver in relation to mood; the apparent contradiction that antidepressant-like effects are accompanied by increased Bmal1 expression, while stress resilience is conferred by absence of Bmal1 expression, is explained by the altered responsiveness of Homer1a in these conditions. The fact that both SR10067 and SR1078 lack effect on Homer1a expression and behavior in mPFC CaMK2a-Bmal1KO animals supports the central assumptions of our model (Fig. 7)." This sentence is confusing and needs a more coherent link between the first and second parts.

57. Figure 7, on the left arm, it appears that ketamine was injected before stress-mediated depression. However, in Figure 1, CDM was followed by the ketamine injection. Please correct the figure.

Reviewer #2 (Remarks to the Author):

Garnder et al. have used panel of tools to investigate the role of mPFC molecular clock in development of depressive-like phenotype and rapid antidepressant responses. The manuscript is generally very interesting and topic is important. Methodology is good and conclusions are in line with the data. My most critical comment is about used animal model, and I propose performing at least part of the experiments using more established animal model of depression. Other comments are more general in nature and can be more easily addressed by the authors.

Major points:

- I have difficulties in understanding the relevance of used "animal model of depression". How unpredictable the swim stress was for the animals (potential impact of learning). What happened to animals weight? I assume the temp conditions of the water resulted in hypothermia of the animals? Is the stress only a disruptor of circadian rhythm? I recommend the authors to reproduce at least part of their experiments using more widely established model (e.g. social defeat, CUMS).
- Why sex was not used as a biological variable throughout the ms.? Both males and females were used but now it is not obvious in the main figure which sex the data represent?
- How the doses of drugs, and dosing times, were selected? The authors could also comment on the kinetics of the drugs. Furthermore, please mention that racemic ketamine was used. Also clearly state what was the free-salt dose for all the drugs.
- What might happen to NMDARs (primary target of ketamine) in all experimental conditions?
- Author's views on available antidepressant therapies seems biased. I recommend more neutral approach. For example, although ketamine may produce adverse effects, these are often quite minor. Author's views on the role of circadian system in depression is also biased. Again, a more neutral approach is encouraged.
- Why n-numbers vary? Did the authors use power calculations?
- Please clarify if the animals were euthanized for mRNA and protein analyses (brain tissues) with cervical dislocation without anesthesia.

Minor comments:

- Language is mostly ok, but I recommend checking the grammar by a professional
- Used animal species should be mentioned in title.
- Please clarify for the reader that light on period and lights off period means inactive phase and active phase, respectively, for mice.

- zeitgeber time should be Zeitgeber Time
- star methods should be STAR methods
- What is known about the off-targets of used pharmacological agents (especially SR10067 and SR1078)
- Please open the figure legends so that the reader understands everything what is shown (especially e.g. Fig 1a and d, and similar).
- Whenever possibly please use the same scale in X-axis for similar analyses (e.g. RT-qPCR data). Always start X-axis from 0 (e.g. Fig 2e).
- I consider some of schematic figures with mouse face icons unscientific. Please consider using another approach.

Reviewer #3 (Remarks to the Author):

Gardner et al. describe the effects of modulation of the circadian clock in the medial prefrontal cortex on depression-like behavior and antidepressant response to ketamine in this manuscript. The authors show that targeted suppression of *Bmal1* in the glutamatergic neurons of the mPFC induces depression-like behavior and then examine a series of drugs targeting components of the molecular clock to examine this in a mechanistic detail. There are a number of inconsistencies in the data as outlined below that significantly weaken the manuscript.

In figure 1, the authors show that the chronic despair model (CDM) causes some shifts in expression in the pattern of some clock genes in the mPFC. They also show that ketamine, when doses at ZT6, induces shifts in the expression of clock genes relative to CDM mice. Ketamine treatment appears to “normalize” the rhythm (e.g. *RORa*) for some genes, but in other cases such as for *Cry 1* it takes a gene that was not altered much by CDM and severely shifts its circadian pattern of expression. A key control that is not included is the effect of ketamine on the rhythm in normal mice. Since ketamine induces sleep and it is being administered during their sleep phase, it would also be important to provide this at a different time as well during their wakefulness phase.

The authors examine the effects of a knockdown of *bmal1* in the glutamatergic neurons of the mPFC. As shown in Fig. 2C, *bmal1* mRNA decreases by 50% (AAV CamMK2a-CRE injected into *Bmal1* floxed mice). The functional relevance of the 50% decrease in *Bmal1* mRNA at ZT06 isn't clear as the effect on the circadian rhythm is not shown. Either evaluation of *Bmal1* protein expression at 2 different circadian times (12h apart), or assessment of mRNA expression at multiple times is required to demonstrate functional knock-down. Additionally, the authors refer to this as *Bmal1* ablation and a 50% reduction should not be considered an ablation. A similar issue occurs with *Per2* knockdown as the impact on the entire rhythm is not clear although in this case, *PER2* protein KD is measured and is significant.

It is interesting the alterations in sucrose preference by the REV-ERB agonist seem to be dependent on the phase shift that is only observed in D/D conditions. Other REV-ERB agonists have been shown to completely block the circadian expression of REV-ERB under D/D conditions but have no effect under L/D conditions and have very distinct effects on other clock genes depending on L/D vs. D/D (PMCID: PMC3343186; supplementary figures 9 and 12). With this anhedonia effect noted at a single time point and only under D/D

conditions, it is interesting that there are effects noted in the TST and CDM models. It should be clarified when discussing these results that these were conducted under L/D conditions since it appears that the only assay conducted under D/D conditions was the sucrose preference. Do the authors have an explanation for these differences? The authors should be careful in broadly describing the anhedonic effects of the compound since

The authors claim that Bmal1 KO inhibits the development of depression-like behavior, but several of the drug specificity studies are not consistent with this. For example, in the specificity studies for SR10067, the authors run the CDM and TST test plus or minus SR10067 in the REV-ERBa KO mice and show no effect, but it is interesting that the REV-ERBa KO itself doesn't show a phenotype since it is expected that Bmal1 levels would be increased. Since REV-ERB has basal "constitutive" activity due to the presence of its natural ligand heme, the KO would be expected to drive the opposite response as SR10067. Bmal1 levels should be elevated and drive the opposite phenotypic response.

This is also the case for the RORa KO mice. RORa drives constitutive transcriptional activation of Bmal1 thus loss of RORa should decrease Bmal1 expression and drive a basal phenotype, which it apparently does not.

Neuroinflammation is induced by stress and is correlated with depression and REV-ERB agonists are very effective in suppression of neuroinflammation (PMID: 35668454, PMID: 30792350). REV-ERBa KO mice have enhanced neuroinflammatory response, which would be expected to drive basal stress and potentially lead to depression-like behavior. These data are inconsistent with the authors data and should be discussed.

Minor:

1. In Figure 2, the order of presentation of the panels is not aligned with the panel numbering.
2. mPFC should be spelled out in the title

REVIEWER COMMENTS

Reviewer #1 (Remarks to the Author):

This is an intriguing piece of work that could significantly contribute to our understanding of the link between the circadian clock and rapid antidepressants. The authors argue that BMAL within the CAMK2a neurons of the MPFC is necessary to induce the ketamine-mediated antidepressant effect through the direct induction of the transcription factor HOMER1a, which, in turn, targets plasticity-related genes. The authors propose a mechanism involving the transcriptional regulator RORa, a known BMAL inducer. Along these lines, the authors argue that the inhibition of BMAL1 by its classic transcriptional repressors PER2, CRY2, or REVERBa induces depressive behavior. The experiments are well-designed, and the authors have employed various complex techniques. Therefore, I believe it is worth considering for publication in Nature Communications after major revisions and addressing the concerns raised in this review.

Authors response: We sincerely thank the Reviewer#1 for dedicating time to thoroughly review our article. The positive feedback and valuable suggestions provided have been instrumental in enhancing the interpretation of our data and strengthening the overall quality of the manuscript. We have addressed the Reviewer's concerns by performing new experiments and revising the original version of the manuscript. We believe the revised version now presents a clearer and more robust description and interpretation of our research. Below is the point-by-point response to the Reviewer.

The manuscript has three main and critical weaknesses that need to be appropriately addressed. Since most of the work is based on the assumption of BMAL's role in the antidepressant effects of different drugs, the elimination within the CAMK2a neurons is not convincing (Fig. 2a, b, and Supplementary Fig. 2a). In Figure 1b, the authors should demonstrate the expression of BMAL1 with or without CRE, which would reflect the result in panel C. The merged figure should clearly illustrate a reduction in the colocalization between BMAL1 and CAMK2A in the mice expressing CRE. Additionally, corresponding quantification should be included.

Authors response: We agree with the reviewer that showing specific Cre-dependent ablation of BMAL1 expression is an important control. In the revised version of the manuscript, we now provide evidence that EGFP-Cre nuclear expression specifically matches downregulated/absent BMAL1 labelling, while in control EGFP positive cells, BMAL1 expression was preserved (Fig 3d). However, BMAL1 is also ubiquitously expressed also in astrocytes, creating a high background, which makes quantification of the immunohistochemistry images difficult. Instead, as requested by Reviewer#3, we have performed western blotting analyses of mPFC lysates from 2 different time points ZT06 and ZT18, thus providing quantifiable evidence for efficient downregulation of BMAL1 protein levels (Fig 3e). Moreover, in the Suppl Fig 3a we demonstrate the specificity of AAV-CaMK2a-CreEGFP expression by showing selective targeting in CaMK2a-positive neurons and no AAV expression (leakage) into GFAP labelled astrocytes.

The text is challenging to follow; the language should be clearer, more concise, and precise to be convincing. The discussion section is extensive, somewhat redundant, and requires a better bibliographical analysis to support the model. Therefore, the manuscript would benefit from careful revision

Authors response: We thank the reviewer for this suggestion. The new version of the manuscript has undergone critical revision; specifically, we have aimed to omit redundancies and more carefully select the bibliography in the discussion section.

On this note, the model appears incomplete, lacks sufficient support, and presents some results that seem contradictory due to the absence of thorough analysis, especially concerning the role of BMAL1. For instance, the authors could explore the involvement of other transcription factors contributing to the regulation of BMAL or RORa, which, under certain circumstances (e.g., stress, ketamine, sleep deprivation), may have either antidepressive or depressive effects.

Authors response: These comments from the reviewer have highlighted to us that we must more clearly present and discuss our model, particularly regarding the role of BMAL1. As presented in our updated model in Fig 3I, we propose that Bmal1KO (specifically in mPFC CaMK2a neurons) inhibits both the development of stress-induced depression-like behavior (Fig 3f-h) and the antidepressant effects of ketamine via blocking Homer1a downregulation by stress and Homer1a induction by ketamine. We have previously shown that Homer1a expression in mPFC correlates with the depression-like behavior: Homer1a is downregulated by stress and its induction is necessary for the effects of several different antidepressants (Serchov et al., Neuron 2015; Holz et al., Neuron 2019). Similarly, Sato et al., Neuroscience 2020 have shown that Bmal1KO leads to blunted response of Homer1a to sleep deprivation, while Homer1a baseline expression was unaffected.

Indeed, we have investigated also other transcriptional factors involved in the regulation of Bmal1 and Homer1a, and other clock genes (like Dbp), most importantly CREB. We did not find any significant changes in CREB phosphorylation and expression in the mPFC (data not included, but discussed), unlike Sato et al.; however, they performed their experiments in total Bmal1KO animals using whole brain lysates, thus potential developmental influence of the KO or a brain-region specific effect should be considered in their study. We agree with the reviewer that there are many other transcriptional factors and clock-controlled genes, that might be included in our final model (Fig 8), but to avoid any speculations, we have focused our discussion only on our own data.

Furthermore, integrating acrophase data from different genes into the results and/or the model would provide valuable insights. Additionally, it would be beneficial if the authors discussed the distinctions between depression-like behavior and anxiety, as they observed different effects on these behaviors.

Authors response: The reviewer is right that integration of the acrophase analyses is an essential aspect in the final data interpretation. We have incorporated both acrophase and amplitude data and analyses for all investigated clock genes (Fig 1c, Suppl Fig 1c and Suppl Fig 2a) and have included discussion on these data.

The reviewer is again correct that discussing the different effects on depression- and anxiety-like behavior is an important subject. Though in many animal stress models increased depression-like behavior is accompanied by an anxiogenic phenotype, our CDM mice lack enhanced anxiety-like behavior (Serchov et al., Neuron 2015; Holz et al., Neuron 2019). Interestingly, manipulations of Rev-ERB (KO or SR10067 agonism) lead to opposite effects – KO cause antidepressant and anxiogenic effect and SR10067 has pro-depressant and anxiolytic effects. These aspects are now discussed in the revised version of the manuscript.

Specific points:

Author response: We thank the reviewer for noticing the inaccuracies and confusing sentences. They were all corrected in the revised version of the manuscript. Please find below brief point by point response to all of them:

1. "We found increased oscillation of clock negative regulatory loop" It is somewhat ambiguous. Is the author referring to the transcription of clock genes or the negative regulators like PER or CRY?

Author response: We are indeed referring to Per and Cry clock genes (upregulated by CDM), which are part of the negative regulatory clock loop. We have rephrased this sentence for more clarity.

2. "normalized by rapid antidepressant ketamine" What does it mean to 'normalized' the increase in the oscillation?

Author response: We meant that ketamine counteracts the CDM effects on the molecular clock by decreasing Per and Cry, but increasing RORa expression and shifting the acrophase of Bmal1 towards the control (non-CDM) state. We agree with the reviewer that term "normalize" is not fully appropriate and we have modified our language accordingly.

3. "clockwork via viral Bmal1KO in CaMK2a excitatory neurons blocks both development of depressive-like phenotype and ketamine's effect ." This could be explained more thoroughly because ketamine is an antidepressant, unlike stress-induced depression.

4. "Repetitive swim stress and rapid antidepressant ketamine alter circadian clock gene expression in the mouse mPFC" These are opposing stimuli; therefore, the title is somewhat confusing

Author response: We apologize for the confusing phrases. We have elaborated these sentences and for clarity have now split these data into 2 separate figures and subchapters, respectively (Fig 1 for CDM effect and Fig 2 for ketamine effect), in the revised version of the manuscript

5. "mice (1 week after the induction phase)" What does 'after the induction phase' mean?

Author response: We apologize for the lack of clarity. Our CDM model consist of 2 phases: 1) induction phase: 10 min swimming for 5 days (induction of depression-like behavior) and 2) test phase: FST, TST and SPT conducted 1-3 weeks after the swim-stress protocol, where potential antidepressant treatment administered between the both phases might be tested. The CDM model is now better explained in the revised version.

6. In Table 1b of the supplementary materials, where only P values are provided, it is not clear how the values change. For instance, in the rhythmicity P values, it would be helpful to represent them as -logP values and include a color or bar scale. For comparative values, consider using a color scale to show the direction of the change, in addition to the -logP value.

7. "(except Per1)" Except Per1, In the supplementary table, the Per1 values are similar to those of the other genes; therefore, this statement is not clear

8. In Supplementary Table 1E, it is not clear what the P-value of the amplitude is referring to, nor the CDM effect. Could you please specify which values are being compared?

Author response: We apologize once again for the lack of detail. To improve the clarity, we have inserted the P values for CDM effect (control vs CDM) in the two-way ANOVA into the respective graphs (Fig 1b & 2b). The calculated values of amplitude and acrophase for the selected genes ($p < 0.05$; control vs CDM) are now plotted and presented in Fig 1c and Suppl Fig 2a. The Tables in Suppl Fig 1 and Suppl Fig 2 are now reorganized and fully explained in the figure legends. The P values represent the significance in rhythmicity, CDM effect in two-way ANOVA (control

vs CDM), amplitude and acrophase (extra sum of squares F test, control vs CDM). We also added the exact amplitude and acrophase values for all 3 conditions (control, CDM and CDM+KET) in the Suppl Fig 2a.

9. "including amplification of negative loop" This is not clear. What do the authors mean by "amplification"? Are they referring to increased amplitude or more robust rhythms?

10. "normalization of Bmal1 oscillation" This is somewhat ambiguous. Does this normalization refer to a phase shift in the acrophase?

11. "potentiation of positive element Rora" The same applies to RORA. It would be better to clarify whether there's an increase in amplitude or a strengthening of the rhythms.

Author response: We apologize for the confusing sentences. They have been rephrased with detailed description of the effects on expression, acrophase and/or amplitude in the revised version of the manuscript.

12. "Targeted Bmal1 knockout in the excitatory CaMK2a neurons of mPFC blocks Homer1a modulation by stress and ketamine and inhibits the development of depression-like behavior and the antidepressant response." The title is confusing. It should separate the ketamine and stress responses, as these are antagonistic and do not sound biologically coherent.

13. Supplemental Figure 2b should include the word 'baseline' in the figure title

14. mice (Fig. 2d) " Figure 2d refers to TST or FST?

15. "thus preventing negative clock element upregulation by CDM (Fig. 2i and Supplementary Fig. 2h)." This statement is confusing. What does 'negative clock element' mean? Additionally, as RORA is not a repressor, it should be discussed in a separate sentence.

16. "and decreased the expression of Per1, Per2 and Rev-erba " rather than decreasing to prevent the CDM-mediated induction of the expression of..."

17. "However, the Bmal1 deletion in mPFC CaMK2a neurons inhibited the antidepressant effects of KET (Fig. 2g, h). Or the elimination of BMAL1 induces antidepressant effects.

Author response: We thank the reviewer for the suggestions. We have modified the titles accordingly. Formal Fig 2d now corresponding to revised Fig 3f represents the immobility time during the induction and test phase FST of the CDM model.

18. Fig. 2k This model is confusing. Perhaps it could be split into two parts: one referring to BMAL1 and the other to KET, both of which are inducers of Homer1 and, consequently, associated with antidepressant effects, rather than solely focusing on the effect of eliminating BMAL. Additionally, when mentioning 'Behavioral response,' please specify the type of behavior, whether it pertains to depressive or non-depressive behavior.

Author response: As mentioned above in our model we propose that BMAL1 is involved in both processes and it mediates 1) stress-induced downregulation of Homer1a and 2) Homer1a upregulation by ketamine; and thus BMAL1 modulates the behavioral response in both ways: stress-induction of depression-like behavior and the antidepressant effect of ketamine. In Fig 4g and Fig 8 we then further elaborate the mechanism how Bmal1 is necessary for the stress-mediated downregulation of Homer1a.

19. "transient hypoactive agent" What does 'hypoactive agent' mean? This term is confusing. It would be better to describe the specific effect induced by this agonist.

Author response: We apologize again for lack of clarity. "Hypoactive" is replaced by agent "reducing the activity".

20. "acute or chronic 5 days injection of SR10067" It is unclear whether 'DD' corresponds to acute or 5-day SR10057 injections.

21. "induced anhedonia-like behavior 48h after injection and phase shift of the locomotor activity (Supplementary Fig. 4c, d)." In Supplementary Figure 4A, the schema shows that SR10067 was tested under both LD and DD conditions. However, in Supplementary Figure 4C, there is only one actogram of SR10067 under DD. To enhance clarity and persuasiveness, it would be necessary to include two actograms (one for the vehicle and one for SR10067) covering both LD and DD periods and depicting the days of SR10067 injections).

Authors response: This comment of the reviewer made clear to us that due to a space limit we did not explain these experiments well enough. In the revised version of the manuscript, we have included two pairs of actograms: one pair (Veh and SR10067) for the LD, where the drug causes a temporary suppression/reduction of the locomotor activity in the first active period after injection and a second pair for DD, where SR10067 produces a significant phase shift of activity. Acute or chronic 5 days SR10076 injections are tested both at LD conditions – for more clarity we have inserted now "(at ZT06)" which indicates LD in contrast to CT06 denoting DD condition.

22. "the effects of single" It appears that DD itself induces some anhedonia, as reflected at 74 hours, while at this time, SR no longer has any effects. How can this be explained?

Authors response: The reviewer is right that DD has anhedonic effect. It has been previously reported that chronic exposure to constant darkness produces a depressive-like phenotype (Gonzalez & Aston-Jones PNAS 2008). In our experiment the mice have been kept at least 10 days at DD prior to the injection. This issue is now clearly commented in the revised version of the manuscript.

23. In Figure 4B, is it the Forced Swim Test (FST) or the Tail Suspension Test (TST)?

Authors response: Former Fig 4b now corresponding to revised Fig 5b represents the immobility time during the induction and test phase FST of the CDM model.

24. "Throughout CDM induction, SR10067- treated mice exhibit consistently elevated immobility time compared to vehicle-treated controls, an effect maintained in the test phase FST and TST (Fig. 4b, c)." In the sucrose preference test under DD, the anhedonic effects of SR10067 were lost after 72 hours. However, in the TST (or FST), the depression-like effects last several days. How can this apparent discrepancy be explained?

25. Why is it that in Supplementary Figure D, there is no effect on anhedonia under LD, while in Figure 4D, there is an effect several days after SR10067 injection?

Authors response: These comments from the reviewer have highlighted to us that we must more clearly present and discuss these issues. Our data shows that the REV-ERB agonist SR10067 at LD conditions leads to enhanced and long lasting (at least 1 week) depression-like behaviour

(increased immobility time in FST/TST and decreased sucrose preference) when combined with the swim stress (Fig 5a-d). However, when SR10067 is administered at LD (alone without any swim stress), it did not affect sucrose preference, clock gene expression (mPFC) or phase shift (Suppl Fig. 5b, d & f). Our data are thus consistent with previously reported data on other REV-ERB agonists also showing no effect on clock gene expression in LD (Solt et al. Nature 2012 PMC3343186 - Fig 2e). On the other hand, when SR10067 is given at DD (alone without swim stress), it shifts the locomotor activity and causes a temporary anhedonic effect (Supp Fig. 5c & 5d), which is in agreement with the published data concerning other RevERB agonists which significantly modulate the circadian gene expression in hypothalamus in DD condition (Solt et al. Nature 2012 PMC3343186 - Fig 2c). These distinct effects between LD and DD suggest that the light input into the SCN, significantly modulates the action of the REV-ERB agonists. REV-ERB agonism still displaying a pro-depressive-like effects in LD, but only when combined with a stressful event (swimming). These issues are now discussed in the revised version of the manuscript.

26. "To test the role of mPFC Bmal1 expression in SR10067 action" This sentence is confusing. Something more specific and explicit would be more appropriate. For example, 'To determine whether BMAL1 in the MPFC is required for the induction of SR10067-mediated depression...'

Author response: We thank the reviewer for the suggestion. We have modified this sentence accordingly.

27. "(Fig. 4j)." Since the figure depicting the elimination of BMAL1 in the CaMKIIa neurons (the primary strategy) is already shown in Figure 2a, it is not necessary to repeat the same figure. Otherwise, readers might initially assume it's a different genetic experiment. Instead, the genetic strategy in Figure 2A can be isolated in one panel, while the IntelliCage and CDM, KET, etc., can be presented in Panel 2B. Therefore, in Figure 4J, simply indicating the mouse genotype is sufficient.

Author response: We thank the reviewer for this comment. As suggested, we have split the former Fig 2a into two panels: 3a and 3b. We have completely removed former Fig 4j, thus referring into the text to the genetic model shown in Fig. 3a and the experimental design in Fig. 5a.

28. "while its hypoactive and anxiolytic action were preserved" Please display the same plots for the WT mice treated with SR10067.

Author response: We thank the reviewer for the suggestion. In order to assess the locomotor activity of the WT mice treated with SR10067, we have taken advantage of the automatic measurements in IntelliCage (instead of open field test, which analyses only a 10 min interval). These data are presented as number of corner visits in Suppl Fig 5b (right plot). Although, it has been previously shown that SR10067 has an anxiolytic effect (Banerjee et al., Nat Commun 2014), we have removed the statement "and anxiolytic action were preserved", since we don't have significant effect in Suppl Fig 5g (right panel).

29. "In addition, expression of Homer1a " To provide more evidence (e.g., ChIP assays, eBOX experiments, etc.) regarding the involvement of BMAL in the regulation of HOMER

Author response: We agree with the reviewer that it is important to refer to direct evidence that BMAL1 regulates Homer1. Indeed, such ChIP experiments have been previously conducted by Sato et al. Neuroscience 2020, demonstrating that BMAL1 bind to the Homer1 promoter in the mouse brain. Moreover, the authors show that Homer1a gene expression is unaltered in the absence of BMAL1, and that its immediate early response to SD is BMAL1-dependent.

30. "these results suggest that the KO of Bmal1 expression in excitatory neurons of mPFC is sufficient to block the pro-depressive effects of SR10067. " This conclusion would be better supported by taking a more biological perspective, such as focusing the conclusion on the roles of REVERB and BMAL molecules.

31. "Supplementary Fig. 5e" The referenced figure is not correct; please correct it.

32. "(Supplementary Fig. 5d). " This figure can be placed at the end of Supplementary Figure 5.

Author response: We thank again the reviewer for the remarks. We have modified our discussion, focusing it on the role of both REV-ERB and BMAL1, corrected the reference and changed the Suppl Fig 5 accordingly.

33. "After administration of KET or SR1078 at ZT06, the SWA recorded during NREM sleep in the mPFC was significantly elevated at various points over the subsequent 24h compared to vehicle-treated animals " It could be interesting to determine circadian parameters related to SWA.

Author response: The reviewer is right that principally considered as a marker of the sleep homeostasis, SWA amplitude may also be directly influenced in part by circadian factors (Lazar et al., Neuroimage 2015). However, our cosinor analyses for SWA circadian rhythmicity didn't reach significance for all the groups (KET and SR1078), thus we were not able to make full comparisons (data not shown/excluded due to space limit).

34. Since it has been demonstrated that KET shows long-term effects that last up to 15 days, it would be interesting to determine if the agonist SR1079 also exhibits long-term antidepressant effects.

Author response: The reviewer identifies an important issue about the duration of the antidepressant effects of SR1078. Indeed, acute administration of SR1078 has robust and rapid antidepressant action, however, it only lasts up to 24h after acute administration. However, similarly to Wang et al., ACS Chem Neurosci 2016, we have also tried chronic 2 weeks treatment, that did not produce conclusive results: longer lasting antidepressant effects in FST (up to 72h), but not in TST, associated with unspecific effects on locomotor activity (data not shown). Though the effects are only temporary after acute administration, we are confident that the approach of utilization of clock modulators in treating depression offers a novel therapeutic strategy.

35. "The brain region- and cell type- specific Bmal1KO mouse model used in our study represents a very precise disruption of the circadian clockwork as it is restricted to the CaMK2a excitatory glutamatergic neurons of the mPFC, " Is CaMK2a exclusively expressed in the MPFC? Please provide more convincing evidence or discuss this point

Author response: We thank the reviewer for this comment. CaMK2a is selectively expressed in pyramidal excitatory forebrain neurons (predominantly in cortex and hippocampus). However, our targeted AAV-Cre microinjections in the mPFC achieves the brain-region selectivity of our approach. This issue is now discussed in the revised version of the manuscript.

36. "Notably, it has been recently demonstrated that Homer1a undergoes bimodal regulation by BMAL1" Please provide more specific details

Author response: We apologize for the lack of clarity. We refer here to the Sato et al. Neuroscience 2020, demonstrating that BMAL1 binds to the Homer1 promoter in mouse brain.

Interestingly, circadian Homer1a gene expression is unaltered in the absence of BMAL1, while its immediate early response to sleep deprivation relies on BMAL1. This report is discussed with more detail in the revised version of the manuscript.

37. It is necessary to measure the expression of CREB because the increase in PERs-CRYs and the reduction of HOMER1 cannot be explained solely by the presence of E-box elements.

Author response: We thank the reviewer for the important suggestion. We have indeed investigated the phosphorylation and expression level of the transcriptional factor CREB in the mPFC of WT CDM as well as CaMK2a-Bmal1KO mice. However, we don't find any significant changes in CREB phosphorylation and expression in the mPFC (data not included and discussed in the manuscript, due to space limit). Therefore, we propose the potential involvement of the corticosterone in the stress-induced increase of Per and Cry expression (Woodruff et al., Endocrinology 2016).

38. "Moreover, Homer1a expression appears inversely correlated with that of negative clock elements. " Where is it observed? Or under what conditions?"

Author response: The reviewer is right that this sentence requires more elaboration. We meant that CDM cause downregulation of Homer1a levels and increased expression of Per and Cry in the mPFC, while ketamine increases Homer1a and decreases Per and Cry genes. We removed it from the revised version of the manuscript.

39. "Importantly, our Bmal1KO in excitatory mPFC neurons appeared to block the downregulation of Homer1a in response to CDM stress." Please refer to the corresponding figure

40. "consistent with the aforementioned inverse correlation with Homer1a," Please refer to the corresponding figure

41. "The CaMK2a-Bmal1KO also led to decreased expression of negative loop genes, consistent with the aforementioned inverse correlation with Homer1a , and potentially contributing to stress resilience by preventing the amplification of negative clock elements typically observed after CDM ." This sentence is confusing because it does not mention a link or mechanism between negative loop genes and Homer1a expression.

Author response: We apologize for the apparent mistakes. We have added the missing figure references and corrected and/or removed the confusing sentences.

42. "Simultaneously, a strong increase in the expression of the positive regulatory element Rora was detected in CaMK2a-Bmal1KO mice; " Based on the literature, the author should speculate about a common mechanism for the transcriptional regulation of Homer1a and RORa.

43. "Rora elevation may be a compensatory mechanism in response to the Bmal1KO, as Bering et al. similarly observed significant upregulation of the Clock positive element in their cortical Bmal1KO model" It would be necessary to provide a more detailed mechanism for this assumption

Author response: We thank the reviewer for the suggestion. However, we did not find any references potentially explaining the mechanism of enhanced RORa expression in Bmal1KO animals. The reviewer is right that the compensatory response is pure speculation and we could not provide any mechanism based on the available literature. We have also discussed now that Bmal1KO leads to a general dysregulation of several clock genes/loops in mPFC, which might lead to feedback/forward upregulation of RORa.

44. "We show that the antidepressant effects of KET administration are associated with a striking normalization of the negative loop potentiation observed after CDM. " The comparison between the amplitudes of these genes in the control, CDM, and ketamine groups should be conducted using their relative values and P-values

Author response: We thank the reviewer for the important suggestion. We have conducted the requested analyses and the detailed comparisons of the amplitudes and acrophases, and the p-values are now included in the table of Supplementary Fig. 2a.

45. " In our CaMK2a-Bmal1KO mice, KET had neither an antidepressant effect on behavior, nor did it downregulate negative loop amplitude or upregulate Homer1a expression. " This sentence is confusing. It contains three ideas that should be separated for better understanding.

46. "Despite some inhibition of the response to stress, " What do you mean by 'response to stress'? Are you referring to a depressive effect?

47. "as KET typically reduces immobility time even in naive (non-stressed) animals, an antidepressant-like effect would still be expected. " This sentence is confusing. Can you please clarify or provide more details? an antidepressant-like effect would still be expected. "

48. "modulation of depressive-like behavior" This is ambiguous

49. "bi-directional manner via its regulation of Homer1a responsiveness to CDM (downregulation) or KET (upregulation). " Why do the authors make this statement?

50. "Further, our data show Homer1a induction is again concurrent with clock-mediated antidepressant effect." Please refer to the figure(s) or provide a hypothesis.

Author response: We apologize again for the lack of clarity in our discussion. These sentences have been removed and/or modified in the revised version of the manuscript.

With the term "response to stress" we meant development of depression-like phenotype in response to stress.

Several reports have shown that ketamine has antidepressant-like effects (i.e., a reduction in immobility behavior in the FST and TST) even in naïve (non-stressed) mice. Thus, we would normally expect such an antidepressant response to ketamine in Bmal1KO animals, even though they do not have an elevated depression-like phenotype. We did not see this, leading to our conclusion that the Bmal1KO also blocks the antidepressant-like effects of ketamine.

51. "Coherent with the Rev-erba downregulation seen after both Bmal1KO and Per2KD, we also show a stress-resilient phenotype in Rev-erbaKO mice." Given that Per2 is part of the negative feedback loop of ebox-targeted promoters, how can we explain the downregulation of REVERBa in the Per2KD?

Author response: The reviewer is again right, we cannot provide mechanistic explanation for the downregulation of REV-ERBa expression in Per2KD, based on the current available literature. Considering that this result is obtained only from a single time point, we might speculate that Per2KD might cause a shift of REV-ERBa expression leading to an impression of downregulation. Thus, further analyses on additional time points are needed in order to make firm conclusion. Therefore, we have removed the statement of RevERBa downregulation from the discussion.

52. "Overall, we propose that direct reduction of these negative regulatory elements, PER2 " Elements

can be confused with DNA-binding motifs within promoters; therefore, I suggest referring to them by their actual names, such as transcription factors or transcriptional regulators.

53. "This interruption of effect is specific to the mPFC: although REV-ERB activity is linked to other mechanisms implicated in depression, including neurogenesis in the hippocampus, dopaminergic system and stress-responsive glucocorticoid systems⁶⁰⁻⁶², the blunted effect in our CaMK2a-Bmal1KO mice confirms again the specific importance of Bmal1 function in excitatory neurons for mood modulation." This sentence is confusing. Initially, the author highlights the importance of REVERB in depression and brain processes such as neurogenesis. However, the text then shifts to discussing BMAL without establishing a clear link between these two proteins.

54. "Similarly, lack of SR10067 effect in Rev-erb α KO mice confirms that the pro-depressive effects are not due to other off-target interactions of SR10067." Perhaps something like the absence of SR10067-induced pro-depressive effects would be more appropriate...

55. The induction of Homer1a is the critical driver in relation to mood; the apparent contradiction that antidepressant-like effects are accompanied by increased Bmal1 expression, while stress resilience is conferred by absence of Bmal1 expression" Please refer to figures.

56. "The induction of Homer1a is the critical driver in relation to mood; the apparent contradiction that antidepressant-like effects are accompanied by increased Bmal1 expression, while stress resilience is conferred by absence of Bmal1 expression, is explained by the altered responsiveness of Homer1a in these conditions. The fact that both SR10067 and SR1078 lack effect on Homer1a expression and behavior in mPFC CaMK2a-Bmal1KO animals supports the central assumptions of our model (Fig. 7)." This sentence is confusing and needs a more coherent link between the first and second parts.

Author response: We sincerely thank the reviewer for dedicating time to thoroughly review the discussion section and for all suggestions. We have carefully revised and modified all the commented sentences.

57. Figure 7, on the left arm, it appears that ketamine was injected before stress-mediated depression. However, in Figure 1, CDM was followed by the ketamine injection. Please correct the figure.

Author response: We thank the reviewer for noticing this apparent mistake. The revised Figure 8 is now corrected accordingly.

Reviewer #2 (Remarks to the Author):

Garnder et al. have used panel of tools to investigate the role of mPFC molecular clock in development of depressive-like phenotype and rapid antidepressant responses. The manuscript is generally very interesting and topic is important. Methodology is good and conclusions are in line with the data. My most critical comment is about used animal model, and I propose performing at least part of the experiments using more established animal model of depression. Other comments are more general in nature and can be more easily addressed by the authors.

Authors response: We thank the Reviewer#2 for the positive feedback and valuable suggestions. In response to the Reviewer's request, we have utilized 2 additional animal models. We believe that our new data provide further evidences supporting our hypothesis. Below is the point-by-point response to the Reviewer.

Major points:

•I have difficulties in understanding the relevance of used “animal model of depression”. How unpredictable the swim stress was for the animals (potential impact of learning). What happened to animals weight? I assume the temp conditions of the water resulted in hypothermia of the animals? Is the stress only a disruptor of circadian rhythm? I recommend the authors to reproduce at least part of their experiments using more widely established model (e.g. social defeat, CUMS).

Authors response: The reviewer is questioning the un/predictability of the swim stress and potential impact of learning in the CDM model. We agree that the CDM model has its limitations, like any other model for depression-like behavior/phenotype. However, also other established animal models use predictable stress (chronic restrain/immobilization stress model) and/or learning-linked paradigms (learned helplessness model). The repetitive predictable swim stress used in the CDM model is designed to mimic everyday human unescapable stress, such as daily repetition of a stressful job and familial stresses.

We agree with the reviewer that we cannot exclude mild hypothermia, though the swim time is restricted to only 10 min in 23°C water. However, swim stress not only increases the immobility time in FST, but it also causes an increased immobility time in TST (different test in comparison to swimming) and more importantly it leads to significant reduction of the motivation- and reward-oriented behavior of mice in sucrose preference, all reversible by antidepressant treatment with ketamine, imipramine and fluoxetine and demonstrated by the high tech automatic analyses in the IntelliCage system (Holz et al., *Neuron* 2019; Serchov et al., *Neuropharmacol* 2020; Sun et al., *J Affective Disord* 2021). Moreover, in response to the Reviewer’s request we have now also provided data about the weight of the animals, demonstrating a significant growth reduction in CDM mice (Suppl Fig 1b).

Indeed, the main conclusions in our manuscript involving the CDM model are fully supported by TST and sucrose preference data from IntelliCage (Suppl Fig 1a; Fig 2h; Fig 5d; Fig 6b; the short in vivo effects of Accell siRNA did not allow us to perform SPT in Fig 4).

In summary, the CDM mice develop a chronic state (up to 4 weeks) of depression-like phenotype (TST, FST and sucrose preference) and the CDM model allows reliable prediction of antidepressant-like effects of several different antidepressant treatments. The model is widely accepted in the literature (Sun et al., *J Neurosci* 2011; Serchov et al., *Neuron* 2015; Hellwig et al., *Brain Behav Immun* 2016; Normann et al., *Biol Psych* 2017; Holz et al. *Neuron* 2019; Serchov et al., *Neuropharmacol* 2020; Sun et al., *J Affect Disorder*; Vestring et al., *JoVe* 2022).

In contrast, to other animal models of depression, such as CUS or chronic mild stress, that utilize multiple stressors directly targeting the circadian rhythm and central clock, the clock gene expression in the SCN and rhythmicity of the CDM mice were not changed. Thus, CDM paradigm selectively highlights the importance of extra-SCN clockwork in key depression-relevant regions such as the PFC in stress response and the subsequent development of depressive phenotypes.

In response to the Reviewer’s request, we have utilized 2 additional established animal models of depression – the chronic corticosterone (CORT) model (Moda-Sava et al., *Science* 2019) (samples obtained in collaboration with Dr. Tomi Rantamaki, Helsinki University) and the learned helplessness (LH) model (samples obtained in collaboration with Dr Michel Barrot group at INCI, Strasbourg). We did not select CUMS and social defeat stress (SDS) models as suggested, as the CUMS paradigm contains stressors directly targeting/dysregulating the circadian rhythm and central clock (for example modulation of light/dark cycle) and SDS is an established model mainly restricted to male mice. In LH mice, our data demonstrate significant increases in expression of the clock negative loop genes *Per1* & *Rev-erba* expression and downregulation of *Homer1a*, as well as decreased expression of positive clock regulators *Bmal1* and *RORa* in the mPFC of LH mice, thus supporting our conclusions and data obtained from the CDM model. Our clock gene expression analyses in the mPFC of CORT model mice did not show significant changes. However, the SPT evaluation of the CORT model mice exhibit large variation and did not show significant decrease in the sucrose preference.

•Why sex was not used as a biological variable throughout the ms.? Both males and females were used but now it is not obvious in the main figure which sex the data represent?

Authors response: The reviewer rises an important point regarding the sex of the animals as biological variable. Indeed, where possible we have used both females and males in the majority of our experiments and we examined sex contribution through our principal analysis. However, as sex was not significantly associated with changes in behavior and/or clock gene expression (Chun et al., 2015 J Biol Rhythm), its effect was not adjusted in our final statistical models. Therefore, in the main figures, we have presented combined data from both males and females.

•How the doses of drugs, and dosing times, were selected? The authors could also comment on the kinetics of the drugs. Furthermore, please mention that racemic ketamine was used. Also clearly state what was the free-salt dose for all the drugs.

Authors response: The reviewer identifies the important issues of dose, kinetics and application time of the used clock modulators. The kinetics were indeed not discussed in the manuscript (due to a space limit), since both drugs have been previously well described:

The synthetic Rev-ERB agonist SR10067 was described and fully evaluated by Banerjee et al., Nat Commun 2016. SR10067 showed a better potency than other Rev-ERB agonists (SR9011 and SR9009): for REV-ERBa IC50: 170nM and for REV-ERBb IC50: 160nM. Assessment of plasma and brain concentrations of SR10067 1h and 6h after i.p. injection (30 mg/kg) has revealed that levels of the compound remain above the IC50 for the receptor 6h after administration. Moreover, SR10067 as well as other Rev-ERB agonist (SR9011 and SR9009) have been effectively used in previous studies at ZT/CT06 (Solt et al., Nature 2012; Banerjee et al., Nat Commun 2016; Amador et al., PLOS One 2016 & Biochem Pharmacol 2018). Therefore, we have selected the same dose (30 mg/kg) and the same dosing time (ZT/CT06) for our experiments.

The synthetic ROR α /g agonist SR1078 has been also previously described (Wang et al., ACS Chem Biol 2010) and in vivo evaluated (Wang et al. ACS Chem Neurosci 2015). The authors show that the brain levels of SR1078 1h after injection are ~4 μ M and are maintained above 1 μ M for at least 8h post administration (10 mg/kg i.p.). Thus, we have used the same dose and dosage time (ZT06) as previously shown to be effective (Wang et al. ACS Chem Neurosci 2015).

We have added in the methods section that racemic mixture of R- and S-ketamine was used for all experiments. The doses of all drugs used are indicated as free-salt.

•What might happen to NMDARs (primary target of ketamine) in all experimental conditions?

Authors response: The dominating hypothesis for the mechanism of action of ketamine postulates that ketamine targets NMDARs located on inhibitory (GABAergic neurons), thus leading to disinhibition. Since our model is primary focused on the molecular clock in CaMK2a-positive excitatory neurons, we have not investigated the effects on NMDARs. Moreover, we did not find any references on how circadian clock modulation affects NMDARs in mPFC.

•Author's views on available antidepressant therapies seems biased. I recommend more neutral approach. For example, although ketamine may produce adverse effects, these are often quite minor. Author's views on the role of circadian system in depression is also biased. Again, a more neutral approach is encouraged.

Authors response: We thank the reviewer for noticing these issues and for the recommendation. We have modified now the introduction and moderated our language in attempt to use more neutral approach in the revised version of the manuscript.

- Why n-numbers vary? Did the authors use power calculations?

Authors response: This comment of the reviewer made clear to us that we did not provide detailed information about the sample size calculation. Sample sizes were determined on the basis of extensive laboratory experience and were verified via power analysis. The parameters used were: two-tailed significance level of 5% ($p < 0.05$), power 80% and effect size 1.33 for the behavioral experiments ($n=10$) and effect size of 2.00 for the molecular biology (qPCR and WB) experiments ($n=5$). The effect size was further adjusted based on our previous experience/expectance and depending on the experimental conditions and analyses in the following experiments: Figs 5g & 5h: $n=6$ previous experimental experience with KET; Figs 5j & 5k & 6g: $n=7$ previous experimental experience with BMAL1KO (Fig 3) and SR drugs (Figs 5 & 6); Figs 5m-o: $n=8$ expected effect size (Otsuka et al., 2022); Fig 6k: $n=7$ previous experience with siHomer1a (Serchov et al., 2015 & Holz et al., 2019). In the following experiments/figures the deviating n-numbers are result of: Fig 3f: compromised videoanalyses; Figs 3g & 3i: climbers; Fig 3j-k: compromised RNA prep; Fig 4: mice are excluded due to surgery inaccuracy; Fig 5m-n: 2 KO animals had seizures during FST; Fig 6b: lost/not functional RFID transponder; Fig 6j: compromised RNA prep; 7d-e n number is varying depending on the signal quality and amount of SWS(REM)/time point

- Please clarify if the animals were euthanized for mRNA and protein analyses (brain tissues) with cervical dislocation without anesthesia.

Authors response: We confirm that all mice were euthanized for the mRNA and protein analyses with cervical dislocation without anesthesia. This information is now added in the methods section of the revised manuscript.

Minor comments:

- Language is mostly ok, but I recommend checking the grammar by a professional

Authors response: Though one of the first authors is native English speaker, the revised version of the manuscript is now extensively language edited.

- Used animal species should be mentioned in title.

Authors response: We thank the reviewer for the important suggestion. We added now the animal species in the title.

- Please clarify for the reader that light on period and lights off period means inactive phase and active phase, respectively, for mice.

- zeitgeber time should be Zeitgeber Time

- star methods should be STAR methods

Authors response: We thank the reviewer for indicating these inaccuracies. All of them are now corrected in the revised version of the manuscript.

- What is known about the off-targets of used pharmacological agents (especially SR10067 and SR1078)

Authors response: The discovery and pharmacokinetics of the synthetic ROR agonist, SR1078, with mixed ROR α / γ activity has been previously described (Wang et al., ACS Chem Biol 2010). SR1078 directly selectively binds to the ligand binding domain of ROR α and ROR γ , and dose-dependently increases the transcriptional activity of these receptors, leading to stimulation of ROR α / γ target gene transcription (Wang et al., ACS Chem Biol 2010; Wang et al., JBC 2010; Wang et al., PLoS One 2012). The authors do not describe any unspecific/off targets/cross reactivity effects on LXR α , LXR β and FXR receptors. SR1078 also functions in vivo effectively increasing the expression of ROR α / γ target genes (Wang et al., ACS Chem Biol 2010; Wang et al. ACS Chem Neurosci 2015).

The SR10067 has been described as highly potent synthetic REV-ERB agonist with a lack of activity (unspecific, off target effects) on a wide range of other targets. They demonstrated that SR10067 displayed no significant activity at any other nuclear receptor or a range of other receptors, ion channels and transporters assessed in the NIMH Psychoactive Drug Screening Program (Banerjee et al., Nat Commun 2016).

Due to a space limits, we added the above information in the methods section.

Moreover, we also tested the specificity action of both drugs on depression-like behavior in RevERBaKO and RORaKO animals respectively (Suppl Figs 5j-n & 6d) showing no off-target effects.

•Please open the figure legends so that the reader understands everything what is shown (especially e.g. Fig 1a and d, and similar).

•Whenever possibly please use the same scale in X-axis for similar analyses (e.g. RT-qPCR data). Always start X-axis from 0 (e.g. Fig 2e).

•I consider some of schematic figures with mouse face icons unscientific. Please consider using another approach.

Authors response: We thank the reviewer for pointing out all these issues. We have expanded the figure legends as much as the word count limits allows, including more detailed information. We also adjusted the start of X-axis to 0 of all SPT graphs (Suppl Fig 1a, Fig 3h, Fig 5d and Fig 6b). Although, we tried to unify the scale in X-axis for the most of RT-qPCR data, it was not possible for all the graphs, since the different genes show completely different amplitudes of rhythmic expression. All mouse face icons were removed from the revised version of the figures.

Reviewer #3 (Remarks to the Author):

Gardner et al. describe the effects of modulation of the circadian clock in the medial prefrontal cortex on depression-like behavior and antidepressant response to ketamine in this manuscript. The authors show that targeted suppression of Bmal1 in the glutamatergic neurons of the mPFC induces depression-like behavior and then examine a series of drugs targeting components of the molecular clock to examine this in a mechanistic detail. There are a number of inconsistencies in the data as outlined below that significantly weaken the manuscript.

Authors response: We thank the Reviewer#3 for the feedback and important suggestions. In order to address the Reviewer's concerns for inconsistency, we have performed new experiments and improved the presentation and discussion of the original manuscript. We believe the revised version now presents a clearer and better description and interpretation of our data. Below is the point-by-point response to the Reviewer.

In figure 1, the authors show that the chronic despair model (CDM) causes some shifts in expression

in the pattern of some clock genes in the mPFC. They also show that ketamine, when doses at ZT6, induces shifts in the expression of clock genes relative to CDM mice. Ketamine treatment appears to “normalize” the rhythm (e.g. RORa) for some genes, but in other cases such as for Cry 1 it takes a gene that was not altered much by CDM and severely shifts its circadian pattern of expression. A key control that is not included is the effect of ketamine on the rhythm in normal mice. Since ketamine induces sleep and it is being administered during their sleep phase, it would also be important to provide this at a different time as well during their wakefulness phase.

Authors response: This comment of the reviewer made clear to us that we did not describe and discuss some of our experiments in enough detail. Indeed, we show that CDM causes significant changes in the amplitude and/or acrophase of expression of the majority of the investigated clock genes in the mPFC: significantly enhanced amplitude of Per2 and Cry2 and decreased amplitude and/or expression of RORa & RORb (Fig 1b, 1c & Suppl Fig 1c), and also shifted acrophase of BMAL1 and Rev-ERBa. In contrast, ketamine application at ZT00, downregulates and shifts the expression of several genes of the negative regulatory clock loop (Per1,2, Cry1,2), upregulates positive clock regulator RORa and shifts BMAL1 (Fig 2b; Suppl Fig 2a). We agree with the reviewer that the term “normalize” is not fully appropriate, thus we have modified our language when describing and discussing these data.

The reviewer is again right that testing the effect of ketamine in normal/naïve mice is an important control. As requested by the reviewer, we have performed these experiments and our data show that ketamine, injected at ZT00, has similar effect on the clock gene expression in the mPFC of naïve mice compared to ketamine at ZT00 in CDM mice: significant downregulation of clock suppressors Per1, Per2, Cry1, Cry2 and Rev-ERBa expression and upregulation of the positive regulator RORa levels (Fig 2c).

The reviewer rises an important issue of the timing of antidepressant administration and testing, since it might potentially disturb the sleep physiology. Indeed, a recent detailed review of the literature on antidepressant research revealed that the majority of the preclinical research reports (published in reputable journals) did not disclose the time of application/testing and that all studies, that did report it, were conducted during the animals resting phase (Alitalo et al., *Progr Neurobiol* 2021).

In all our clock gene mRNA analyses in WT CDM mice, as well as on mPFC-CaMK2a-BMAL1KO mice, ketamine was applied at ZT00 (now clearly indicated in newly created Fig2a and Fig3b). We injected ketamine at ZT06 only in our EEG/LFP experiments, in order to be comparable with the SR1078 application (Fig 7c-e). The logic behind our approach is the finding of Bellet et al., 2011 that ketamine directly influences CLOCK/BMAL1 function in time dependant manner and considering that BMAL1 protein expression and activity in the rodent brain peaks at early-midday (ZT00-ZT06, slightly shifted from RNA peak of expression).

As requested by the reviewer, we tested the effects of ketamine administration at ZT12 (beginning of active phase) on the clock genes expression in the mPFC of naïve mice. Our results show similarities to the ZT00 KET application – significant downregulation of clock suppressors Cry1 and Cry2 and increase of positive clock regulator RORa levels, however, we did not find any ketamine-mediated effects on Per1, Per2 and RevERBa expression (Suppl Fig 2d). Taken together our data and others (Alitao et al., 2021) indicate that the timing of ketamine administration might play a role for the clock modulation, but also suggest that it might be also a factor for the magnitude and duration of the following antidepressant effect. We believe that these findings form the basis of a potentially new project (investigating in detail the effects of different time points of ketamine administration on the clock genes, behavior and plasticity), distinct from the main focus of this manuscript.

The authors examine the effects of a knockdown of bmal1 in the glutaminergic neurons of the mPFC. As shown in Fig. 2C, bmal1 mRNA decreases by 50% (AAV CamMK2a-CRE injected into Bmal1 floxed mice). The functional relevance of the 50% decrease in Bmal1 mRNA at ZT06 isn't clear as the effect

on the circadian rhythm is not shown. Either evaluation of Bmal1 protein expression at 2 different circadian times (12h apart), or assessment of mRNA expression at multiple times is required to demonstrate functional knock-down. Additionally, the authors refer to this as Bmal1 ablation and a 50% reduction should not be considered an ablation. A similar issue occurs with Per2 knockdown as the impact on the entire rhythm is not clear although in this case, PER2 protein KD is measured and is significant.

Authors response: We thank the reviewer for this suggestion. Providing evidence of downregulation of BMAL1 protein expression is an important control. In the revised version of the manuscript, we now show significant decrease of BMAL1 protein levels at 2 different time points (ZT06 and ZT18) (Fig 3e). Indeed, we have achieved only about 50% reduction of the RNA/protein expression in the mPFC, which might be a result of the cell-type specificity of our approach: AAV is targeting only CaMK2a-positive glutamatergic neurons, while the rest of the cell populations (inhibitory neurons and glial cells) are still expressing BMAL1 (Fig 3d and Suppl Fig 3a). This issue is now discussed in the revised version of the manuscript.

It is interesting the alterations in sucrose preference by the REV-ERB agonist seem to be dependent on the phase shift that is only observed in D/D conditions. Other REV-ERB agonists have been shown to completely block the circadian expression of REV-ERB under D/D conditions but have no effect under L/D conditions and have very distinct effects on other clock genes depending on L/D vs. D/D (PMCID: PMC3343186; supplementary figures 9 and 12). With this anhedonia effect noted at a single time point and only under D/D conditions, it is interesting that there are effects noted in the TST and CDM models. It should be clarified when discussing these results that these were conducted under L/D conditions since it appears that the only assay conducted under D/D conditions was the sucrose preference. Do the authors have an explanation for these differences? The authors should be careful in broadly describing the anhedonic effects of the compound since.

Authors response: These comments from the reviewer have highlighted to us that we must more clearly present and discuss these issues. Our data shows that the REV-ERB agonist SR10067 at LD conditions leads to enhanced and long lasting (at least 1 week) depression-like behaviour (increased immobility time in FST/TST and decreased sucrose preference) when combined with the swim stress (Fig 5a-d). However, when SR10067 is administered at LD alone (without any swim stress), it did not affect sucrose preference, clock gene expression (mPFC) or cause phase shift (Suppl Fig. 5b, d & f). Our data are thus consistent with other REV-ERB agonists also showing no effect on clock genes expression in LD (Solt et al. Nature 2012 PMC3343186 - Fig 2e). Conversely, when SR10067 is given at DD (alone without swim stress), it shifts the locomotor activity and causes a temporary anhedonic effect (Supp Fig. 5c & 5d), which correlates with the data of other RevERB agonists at DD significantly modulating the circadian gene expression in hypothalamus (Solt et al. Nature 2012 PMC3343186 - Fig 2c). These distinct effects between LD and DD suggest that the light input into the SCN, significantly modulates the action of REV-ERB agonists. Though, REV-ERB agonism is still acting pro-depressive-like at LD only when combined with stressful event (swimming). These issues are now discussed in detail in the revised version of the manuscript.

The authors claim that Bmal1 KO inhibits the development of depression-like behavior, but several of the drug specificity studies are not consistent with this. For example, in the specificity studies for SR10067, the authors run the CDM and TST test plus or minus SR10067 in the REV-ERBa KO mice and show no effect, but it is interesting that the REV-ERBa KO itself doesn't show a phenotype since it is expected that Bmal1 levels would be increased. Since REV-ERB has basal "constitutive" activity due to the presence of its natural ligand heme, the KO would be expected to drive the opposite response as SR10067. Bmal1 levels should be elevated and drive the opposite phenotypic response.

Authors response: We apologize once again for the lack of clarity in presenting and discussing our results and conclusions and thus leading to interpretation of inconsistency. As presented in our updated model in Fig 3l, we propose that Bmal1KO (specifically in mPFC CaMK2a neurons) inhibits both the development of depression-like behavior (Fig 3f-h) and the antidepressant effects of ketamine via blocking Homer1a modulation by stress (downregulation) and ketamine (induction). We have been previously shown that Homer1a expression in mPFC correlates with the depression-like behavior – it is downregulated by stress and its induction is necessary for the effects of several different antidepressants (Serchov et al., Neuron 2015; Holz et al., Neuron 2019). Similarly, Sato et al., Neuroscience 2020 have shown that Bmal1KO leads to blunted response of Homer1a to sleep deprivation, while Homer1a baseline expression was not affected. We hypothesize that BMAL1 might attract and/or interact with other potential upstream regulators of Homer1a, including clock suppressors genes (like Per2; enhanced by stress – CDM & LH, as shown here) or potentiators (like GSK3b; activated by ketamine) and thus modulate the behavioral phenotype. Thus, SR1078-treated and Rev-ERBaKO mice have enhanced BMAL1, leading to increased Homer1a expression in mPFC (Fig 5o) and resistance towards development of depression-like phenotype after repetitive swim stress in our CDM paradigm (Fig 5m & 5n; Suppl Fig 5k & 5l). These issues are now discussed in the revised version of the manuscript.

This is also the case for the RORa KO mice. RORa drives constitutive transcriptional activation of Bmal1 thus loss of RORa should decrease Bmal1 expression and drive a basal phenotype, which it apparently does not.

Authors response: The reviewer is right that if ROR agonist SR1078 increases Bmal1 and Homer1a expression and has antidepressant effects, then RORalphaKO should decrease BMAL1 expression (shown in Suppl Fig. 6d) and thus downregulate Homer1a (not significant Suppl Fig 6d) and thus leading to a pro-depressant-like phenotype. However, RORalpha deficient mice have been described as showing severe ataxia/staggering phenotype (Sidman et al., Science 1962). Nevertheless, they were very good swimmers, showing almost no immobility time in FST. This complex phenotype made the behavioral evaluation of locomotor activity in open field test and immobility in FST difficult and incomparable to WT mice. Therefore, we have shown and interpret here only the behavioral data (immobility time) of TST performed in the test phase 24h after SR1078 application. Our results show that this SR1078 has no off-target effects on the behavior (depression-like behavior in TST), nor on BMAL1 and Homer1a expression in mPFC of RORaKO mice (Fig Suppl Fig 6d). The critical point of the staggerer phenotype of RORaKO mice is now discussed in the revised version of the manuscript.

Neuroinflammation is induced by stress and is correlated with depression and REV-ERB agonists are very effective in suppression of neuroinflammation (PMID: 35668454, PMID: 30792350). REV-ERBa KO mice have enhanced neuroinflammatory response, which would be expected to drive basal stress and potentially lead to depression-like behavior. These data are inconsistent with the authors data and should be discussed.

Authors response: The reviewer identifies an important issue about the role of neuroinflammation in the pathophysiology and treatment of depression. The reviewer is right that REV-ERBaKO mice have enhanced neuroinflammatory response, which might lead to depression-like behavior. However, our experimental conditions did not represent an inflammatory challenge, which might explain the absence of enhanced depression-like phenotype. Moreover, in support of our data Chung et al., Cell 2014 have shown that pharmacological antagonism of RevERBa with SR8278 produces similar phenotype to REV-

ERBaKO (decreased immobility in TST). These issues are now discussed in the revised version of the manuscript.

Minor:

1. In Figure 2, the order of presentation of the panels is not aligned with the panel numbering.

Authors response: We thank the reviewer for pointing out this obvious mistake and apologize for it. We have now corrected the panel numbering in the revised version of the manuscript.

2. mPFC should be spelled out in the title

Authors response: We agree with the reviewer that the title should not contain any abbreviations. Though, we consider mPFC as a common and acceptable abbreviation in the field, we have modified the title as suggested by the reviewer.

REVIEWER COMMENTS

Reviewer #1 (Remarks to the Author):

The authors have significantly improved the manuscript; however, additional improvements are still needed. Also, I would appreciate one-by-one responses.

1. "Selective disruption of mPFC clockwork via viral Bmal1KO in CaMK2a excitatory neurons blocks both development of stress-induced depression-like phenotype and antidepressant effects of ketamine " This sounds counterintuitive because the elimination of Bmal1 has a protective effect against depression, while at the same time, it blocks the antidepressant effect of ketamine. I believe that the abstract should convey clearer ideas.
2. Images in figure 3C still are confusing, the upper panel in the merged image should display higher colocalization between EGFP and BMAL1 and these points should be signaled with arrow heads. Also, a quantification would be required showing higher colocalization in the EGFP/BMAL1 in the WT vs BMAL1KO. Maybe trying to use and merge different colors to get better colocalization signal. Moreover, the 500um picture seems isolated. Please include two 500um microphotograph with a dotted line square zooming the corresponding area of 20 um micrographs in both mice.
3. It has been reported that Homer1a exhibits circadian expression in the brain. Therefore, it would be important to include the circadian expression of Homer1a and, if possible, some of its target genes from the RNA samples in figures 1 and 2.
4. Line 177 "Ketamine also robustly shifted the phase of several clock genes, thus causing a normalization of Bmal1 acrophase to match control" To support this claim, the authors should compare Ketamine (KET) versus control and include these results in Supplemental Figure 2a.
5. Line 178 "Moreover, the reduced expression of Rora in the CDM mice was markedly upregulated by ketamine" This sentence is confusing. How can reduced expression be upregulated?
6. In Figure 2C, since there are two ZT06 measurements showing different effects in some cases, the authors should include the post-injection time below the ZTs and discuss this further.
7. In Figures 2C and Supplemental Figure 2D, it is unclear whether the differential effects observed in some cases between Ketamine injected at ZT0 versus ZT12 are due to the timing of injection or the post-injection time. For instance, Ketamine has a half-life of 3-4 hours. Therefore, the authors should discuss this point.
8. Line 216 "(control) and CDM mice (Fig. 3j and Supplementary Fig. 3h)." please verify Fig. 3h
9. For better clarity, consider separating the KET effect from plots 3J into a dedicated plot (similar to 3K). Additionally, you could move plot 3i (red bars) to follow 3J. This would visually link the effect of KET on FST in both genotypes, followed by the effect on gene expression. Alternatively, if keeping the current layout, you could use the same color scheme as the other plots for the KET data point in 3i, but with a distinct fill pattern (e.g., diagonal lines)
10. Line 224 "ablated in CaMK2a-Bmal1KO animals (Fig. 3k and Supplementary Fig. 3i)." Supp. 3i is missing.
11. How can we explain the increase in Homer1a expression in the Bmal1KO mice? According to the model presented in Figure 4g, Homer1a expression is positively regulated by BMAL1/CLOCK. Please discuss this discrepancy.
12. Line 246 "Manipulation of REV-ERBa activity and expression modulates depression-like phenotype and Homer1a expression via Bmal1 in mPFC" The title is confusing. It would be

better to use something more concise and biologically coherent, such as 'REV-ERBa Induces Depression-like Behavior through Repression of the Bmal1-Homer1a Axis.

13. This is confusing. It seems that in Figure 5A of the experimental design, it is necessary to specify that the mice are the CaMK2a-Bmal1KO mice. Perhaps the authors should consider including a new experimental design or adjusting the text to clarify, such as 'as in (Fig. 3a) ... as in (Fig. 5a)."

14. Line 276 "these results suggest that the KO of Bmal1 expression in excitatory neurons" This is confusing. Please substitute 'KO' for another term, such as 'deletion' or 'reduction of Bmal1 expression.

15. Figure 5i should be in lowercase letters instead of capital letters

16. Line 281 "showing low immobility" better "reduced immobility time"

17. In Figure 5j, for clarity and greater persuasiveness, consider including the data from Figure 3f in the same plot.

18. Line 292 "Our data demonstrate the effects of pharmacological manipulation of the clock on mood, with an enhancement of stress-induced depression-like behavior and inhibition of ketamine antidepressant action by REV-ERB agonism, mediated via Bmal1 and Homer1a in the mPFC (Fig. 5i)". This sentence is confusing.

19. Why is there only an antidepressant effect under CDM conditions with sr1078? Please discuss this.

20. Line 358 "None of the treatments had a significant effect on the time spent in the different vigilance states (wake, SWS, REM sleep) (Supplementary Fig. 7e-g)." In Figure 7d, the authors demonstrate an increase in SWA NREM in both KET and SR1078. However, it remains unclear why no such effects are shown in Supplementary Figure 7g. Could you please clarify this point?

21. Line 385 "is uniquely vulnerable to dysregulation" what do the authors mean with uniquely vulnerable? Please clarify this in the text.

22. Line 435 "the CaMK2a-Bmal1KO mice exhibited both stress-resilient and antidepressant-resistant behavioral changes, suggesting that Bmal1 expression and clock function in these particular neurons in mPFC may be an important factor in vulnerability to stress and the development and treatment of depression-like behavior¹²." Line 438 Nevertheless, CaMK2a-Bmal1KO mice showed increased immobility and lower sucrose preference after the CDM paradigm" This last sentence contradicts the results shown in Figures 3f, g, h

23. Line 456 "BMAL1 might attract and/or interact with other potential upstream regulators of Homer1a, including clock suppressor genes (enhanced by stress, CDM and/or LH) or potentiators (activated by ketamine)^{63, 64} and consequently modulate the behavioral phenotype" This is confusing.

24. Line 459 "Thus, the apparent contradiction that antidepressant-like effects are accompanied by increased Bmal1 expression (Fig. 5m-o, 6b-e), while stress resilience is conferred by absence of Bmal1 expression (Fig. 3f-h) is explained by the altered responsiveness of Homer1a" Together, these data suggest that a functional mPFC clock and in particular the presence of Bmal1 is necessary for both the development of stress-induced depression-like behavior and the antidepressant-like effects of ketamine via blocking Homer1a downregulation or induction by stress and ketamine, respectively." Since this 'contradictory' observation is significant, further speculation is required. For example, consider whether the negative components inhibit Homer1 expression or the sample timing.

25. Line 472 "As previously reported the distinct effects between LD and DD suggest that the light input into the SCN, significantly modulates the action of the REV-ERB agonists³⁴" What effects are the authors referring to?

26. Line 507 "ketamine and SR1078" rather ketamine or SR1078

Reviewer #3 (Remarks to the Author):

The authors have adequately addressed my concerns. This manuscript has been improved considerably.

Response to Reviewers Comments

Reviewer #1 (Remarks to the Author):

The authors have significantly improved the manuscript; however, additional improvements are still needed. Also, I would appreciate one-by-one responses.

Authors' response: We thank the Reviewer for the positive feedback and the valuable suggestions. Below are the point-by-point responses to the Reviewer's comments.

1. "Selective disruption of mPFC clockwork via viral Bmal1KO in CaMK2a excitatory neurons blocks both development of stress-induced depression-like phenotype and antidepressant effects of ketamine " This sounds counterintuitive because the elimination of Bmal1 has a protective effect against depression, while at the same time, it blocks the antidepressant effect of ketamine. I believe that the abstract should convey clearer ideas.

Authors' response: We apologize for the confusing phrase. We have modified the abstract accordingly, so that it is now clearly conveying our main messages (see yellow highlighted sections). However, the Nat Commun journal format allows only 150 words limit for the Abstract, which restricts the detailed explanations.

2. Images in figure 3C still are confusing, the upper panel in the merged image should display higher colocalization between EGFP and BMAL1 and these points should be signaled with arrow heads. Also, a quantification would be required showing higher colocalization in the EGFP/BMAL1 in the WT vs BMAL1KO. Maybe trying to use and merge different colors to get better colocalization signal. Moreover, the 500um picture seems isolated. Please include two 500um microphotograph with a dotted line square zooming the corresponding area of 20 um micrographs in both mice.

Authors' response: We thank the reviewer for noticing these issues. In the newly revised version of the manuscript, we have replaced the CaMK2a-EGFP/BMAL1 immunofluorescence images with new representative pictures showing higher colocalization between EGFP and BMAL1 stainings – indicated by arrows (Figure 3C). As requested by the reviewer, we also performed quantifications of the EGFP/BMAL1 colocalization and thus demonstrating higher colocalization in the CaMK2a-EGFP in comparison to very low values in CaMK2a-Cre expressing cells (Suppl Fig. 3B). Please note that the CaMK2a-EGFP-Cre expression is mainly found in the nucleus (due to the nuclear localization signal of EGFP-Cre conjugate), while CaMK2a-EGFP expression is located in the whole cell body. In addition, we also added a second 500um image and indicated the corresponding area of 20um micrographs (Figure 3c).

3. It has been reported that Homer1a exhibits circadian expression in the brain. Therefore, it would be important to include the circadian expression of Homer1a and, if possible, some of its target genes from the RNA samples in figures 1 and 2.

Authors' response: The reviewer is right that several reports have demonstrated oscillating Homer1a expression in the brain (Nelson et al., Neurosci Lett 2004; Maret et al., PNAS 2007, Sato et al., Neurosci 2020). As requested, we have included Homer1a expression profile in the mPFC (Suppl Fig 3f). Indeed, our data show a similar oscillating expression pattern, with peak expression at ZT18, as previously reported.

In contrast to the canonical clock genes that act as transcriptional factors/regulators, Homer proteins are postsynaptic scaffolding proteins that form multimeric complexes, linking several postsynaptic receptors (mGluR5, AMPARec, NMDARec). Homer1a is a shorter isoform rapidly regulated by neuronal activity, that lacks the dimerization domain, and thus it reorganizes the postsynaptic scaffolds and modulates the functioning of these receptors (activity and trafficking). Therefore, we have investigated AMPARec synaptic expression and SWA (delta

power) as potential downstream effects of Homer1a action. Thus, the clock genes investigated in Fig 1 and Fig 2 are not target genes of Homer1a, but rather its potential regulators, particularly BMAL1 and Per2, as proposed in Fig3 and Fig4.

4. Line 177 "Ketamine also robustly shifted the phase of several clock genes, thus causing a normalization of Bmal1 acrophase to match control" To support this claim, the authors should compare Ketamine (KET) versus control and include these results in Supplemental Figure 2a.

Authors' response: The reviewer is again correct that in order to support our claim, a direct comparison between control and KET is necessary. We have now performed these analyses and included the statistical data in Suppl. Fig 2a.

5. Line 178 "Moreover, the reduced expression of Ror α in the CDM mice was markedly upregulated by ketamine" This sentence is confusing. How can reduced expression be upregulated?

Authors' response: We thank the reviewer for noticing the confusing phrase and apologize for it. We have now corrected it accordingly (see yellow highlighted sections).

6. In Figure 2C, since there are two ZT06 measurements showing different effects in some cases, the authors should include the post-injection time below the ZTs and discuss this further.

Authors' response: We agree with the reviewer that it is appropriate to include the ketamine post-injection time below the ZTs. We have therefore added that information in all data plots showing ketamine effects – including Figure 2b, 2c and Suppl Fig 2d.

7. In Figures 2C and Supplemental Figure 2D, it is unclear whether the differential effects observed in some cases between Ketamine injected at ZT0 versus ZT12 are due to the timing of injection or the post-injection time. For instance, Ketamine has a half-life of 3-4 hours. Therefore, the authors should discuss this point.

Authors' response: Indeed, the effects of ketamine injected at ZT00 and ZT12 differ. When injected at ZT00 ketamine downregulates all investigated negative clock regulators: Per1, Per2, Cry1, Cry2 and RevERBa, while at ZT12 it decreases only the expression of Cry1 and Cry2. Bellet et al. have previously demonstrated that ketamine directly alters CLOCK/BMAL1 promoter recruitment in a time-dependent manner and thus modulates clock gene transcription (Bellet et al., PlosONE 2011). Therefore, the stronger effects of ketamine on the clock gene expression when applied at ZT00 vs ZT12 might be explained by the higher expression and activity of BMAL1 during the early day (ZT00-ZT06), compared to early night. Thus, the timing of ketamine treatment may play a role in clock modulation and thus be also a factor for the magnitude and duration of the subsequent antidepressant effects. These issues are now included in the discussion section of the newly revised version of the manuscript (see yellow highlighted sections).

8. Line 216 "(control) and CDM mice (Fig. 3j and Supplementary Fig. 3h)." please verify Fig. 3h

Authors' response: We thank the reviewer for noticing this mistake and apologize for it. The incorrect figure references have been corrected (see yellow highlighted sections).

9. For better clarity, consider separating the KET effect from plots 3J into a dedicated plot (similar to 3K). Additionally, you could move plot 3i (red bars) to follow 3J. This would visually link the effect of KET on FST in both genotypes, followed by the effect on gene expression. Alternatively, if keeping the current layout, you could use the same color scheme as the other plots for the KET data point in 3i, but with a distinct fill pattern (e.g., diagonal lines)

Authors' response: Following the reviewer suggestions, we have reorganized Figure 3, replacing the order of panels i and j. Moreover, for better clarity and consistency, we have replaced the white/red bars with green/black colors for CaMK2aEGFP and CRE, respectively.

10. Line 224 "ablated in CaMK2a-Bmal1KO animals (Fig. 3k and Supplementary Fig. 3i)." Supp. 3i is missing.

Authors' response: We thank the reviewer for noticing this mistake and apologize for it. The incorrect figure references have been fixed (see yellow highlighted sections).

11. How can we explain the increase in Homer1a expression in the Bmal1KO mice? According to the model presented in Figure 4g, Homer1a expression is positively regulated by BMAL1/CLOCK. Please discuss this discrepancy.

Authors' response: This comment of the reviewer made clear to us that due to a space limit, we did not explain these experiments well enough. We did not observe any increase in Homer1a expression in the Bmal1KO mice (see Figure 3i, last panel). Homer1a expression is downregulated by the CDM (as previously shown in control mice: Sun et al., J Neurosci 2011; Serchov et al., Neuron 2015; Holz et al., Neuron 2019; Serchov et al., Neuropharmacol 2020; Sun et al., J Affect Disord 2021) in the CaMK2a-EGFP mice, while it is not changed in CaMK2a-Bmal1KO mice (Homer1a modulation by CDM and/or ketamine is suppressed in CaMK2a-Bmal1KO).

12. Line 246 "Manipulation of REV-ERB α activity and expression modulates depression-like phenotype and Homer1a expression via Bmal1 in mPFC" The title is confusing. It would be better to use something more concise and biologically coherent, such as 'REV-ERB α Induces Depression-like Behavior through Repression of the Bmal1-Homer1a Axis.

Authors' response: We thank the reviewer for the suggestion. We have modified the confusing title (see yellow highlighted sections).

13. This is confusing. It seems that in Figure 5A of the experimental design, it is necessary to specify that the mice are the CaMK2a-Bmal1KO mice. Perhaps the authors should consider including a new experimental design or adjusting the text to clarify, such as 'as in (Fig. 3a) ... as in (Fig. 5a)."

Authors' response: We thank the reviewer for the suggestion. We have now included a new experimental design schema (Fig 5j) for the experiments in Fig 5k-m.

14. Line 276 "these results suggest that the KO of Bmal1 expression in excitatory neurons" This is confusing. Please substitute 'KO' for another term, such as 'deletion' or 'reduction of Bmal1 expression.

Authors' response: As suggested by the reviewer, we have replaced "KO" with "deletion" (see yellow highlighted sections)

15. Figure 5i should be in lowercase letters instead of capital letters

Authors' response: We thank the reviewer for noticing this mistake and apologize for it. The capital letter in Fig 5i has been corrected.

16. Line 281 "showing low immobility" better "reduced immobility time"

Authors' response: As suggested by the reviewer, we have replaced “low” with “reduced” (see yellow highlighted sections).

17. In Figure 5j, for clarity and greater persuasiveness, consider including the data from Figure 3f in the same plot.

Authors' response: We thank the reviewer for the suggestion. As requested, we have combined the data from Fig3f and Fig 5j and presented it as a separate plot in Suppl. Fig 5g.

18. Line 292 “Our data demonstrate the effects of pharmacological manipulation of the clock on mood, with an enhancement of stress-induced depression-like behavior and inhibition of ketamine antidepressant action by REV-ERB agonism, mediated via Bmal1 and Homer1a in the mPFC (Fig. 5i)”. This sentence is confusing.

Authors' response: We apologize for the confusing sentence. We have modified it in the newly revised version of the manuscript (see yellow highlighted sections).

19. Why is there only an antidepressant effect under CDM conditions with sr1078? Please discuss this.

Authors' response: The reviewer challenges the question of the lack of effects of SR1078 in control naïve mice. Indeed, SR1078 promotes antidepressant effects only in the CDM model – mice with already induced depression-like phenotype, thus showing specific antidepressant potential. Any antidepressant-like action of SR1078 in naïve mice might be interpreted as an unspecific and/or manic effect on behavior. Another plausible explanation for the lack of SR1078 effects in naïve mice, might be that SR1078 influences specifically only the dysregulated clock of the CDM model. We have briefly discussed this point in the newly revised version of the manuscript (see yellow highlighted sections).

20. Line 358 “None of the treatments had a significant effect on the time spent in the different vigilance states (wake, SWS, REM sleep) (Supplementary Fig. 7e-g).” In Figure 7d, the authors demonstrate an increase in SWA NREM in both KET and SR1078. However, it remains unclear why no such effects are shown in Supplementary Figure 7g. Could you please clarify this point?

Authors' response: The reviewer is right, SR1078 and KET had no significant effect on the time spent in the different vigilance states: wake, SWS(NREM) and REM sleep (Suppl Fig 7e-g). However, both SR1078 and KET significantly increased the delta power/slow wave activity during NREM (SWS) recorded specifically by local field potentials (LFP) in mPFC (Fig 7d). In addition, we also observed only a transient increase of the delta power/slow wave activity during NREM(SWS) recorded by global ECoG (Fig 7e). In order, to increase the clarity, we have modified the titles of Fig 7d and 7e.

21. Line 385 “is uniquely vulnerable to dysregulation” what do the authors mean with uniquely vulnerable? Please clarify this in the text.

Authors' response: We thank the reviewer for noticing this mistake and apologize for it. We have removed the word “uniquely” from the sentence.

22. Line 435 “the CaMK2a-Bmal1KO mice exhibited both stress-resilient and antidepressant-resistant behavioral changes, suggesting that Bmal1 expression and clock function in these particular neurons in mPFC may be an important factor in vulnerability to stress and the development and treatment of depression-like behavior¹².” Line 438 “Nevertheless, CaMK2a-Bmal1KO mice showed increased immobility and lower sucrose preference after the CDM paradigm” This last sentence contradicts the results shown in Figures 3f, g, h

Authors' response: We agree with the reviewer that the sentence is contradictory. Therefore, we have removed it in the newly revised version of the manuscript.

23. Line 456 "BMAL1 might attract and/or interact with other potential upstream regulators of Homer1a, including clock suppressors genes (enhanced by stress, CDM and/or LH) or potentiators (activated by ketamine)^{63, 64} and consequently modulate the behavioral phenotype" This is confusing.

Authors' response: We apologize for the confusing statement. We are discussing/speculating that BMAL1 is necessary for the regulation of Homer1a expression in both directions (downregulation and induction): 1) potential BMAL1 interaction with clock suppressor genes (Per1,2 and Cry1,2, that are increased by stress) downregulates Homer1a; 2) potential BMAL1 interaction with GSK3b (activated by ketamine) increase Homer1a expression. In order to be more clear, we have expanded this sentence in the discussion (see yellow highlighted sections).

24. Line 459 "Thus, the apparent contradiction that antidepressant-like effects are accompanied by increased Bmal1 expression (Fig. 5m-o, 6b-e), while stress resilience is conferred by absence of Bmal1 expression (Fig. 3f-h) is explained by the altered responsiveness of Homer1a" Together, these data suggest that a functional mPFC clock and in particular the presence of Bmal1 is necessary for both the development of stress-induced depression-like behavior and the antidepressant-like effects of ketamine via blocking Homer1a downregulation or induction by stress and ketamine, respectively." Since this 'contradictory' observation is significant, further speculation is required. For example, consider whether the negative components inhibit Homer1 expression or the sample timing.

Authors' response: We thank the reviewer for the valuable suggestion. Indeed, our data demonstrate that the clock suppressor Per2 (increased by stress in CDM) might negatively regulate Homer1a (downregulated by CDM). Please see the above response.

25. Line 472 "As previously reported the distinct effects between LD and DD suggest that the light input into the SCN, significantly modulates the action of the REV-ERB agonists³⁴" What effects are the authors referring to?

Authors' response: We apologize for the lack of clarity. We are referring to the distinct effects of REV-ERB agonism on clock gene expression in LD vs DD observed by Solt et al, Nature 2012. We have corrected this sentence in the newly revised version of the manuscript (see yellow highlighted sections).

26. Line 507 "ketamine and SR1078" rather ketamine or SR1078

Authors' response: We thank the reviewer for the suggestion. We have corrected this sentence accordingly (see yellow highlighted sections).

Reviewer #3 (Remarks to the Author):

The authors have adequately addressed my concerns. This manuscript has been improved considerably.

Authors response: We thank the Reviewer for the positive feedback

REVIEWERS' COMMENTS

Reviewer #1 (Remarks to the Author):

The authors had responded satisfactorily to my comments